# Learning Diverse Bimanual Dexterous Manipulation Skills from Human Demonstrations

## Abstract

Bimanual dexterous manipulation is a critical yet underexplored area in robotics. Its high-dimensional action space and inherent task complexity present significant challenges for policy learning, and the limited task diversity in existing benchmarks hinders general-purpose skill development. Existing approaches largely depend on reinforcement learning, often constrained by intricately designed reward functions tailored to a narrow set of tasks. In this work, we present a novel approach for efficiently learning diverse bimanual dexterous skills from abundant human demonstrations. Specifically, we introduce **BiDexHD**, a framework that unifies task construction from existing bimanual datasets and employs teacher-student policy learning to address all tasks. The teacher learns state-based policies using a general two-stage reward function across tasks with shared behaviors, while the student distills the learned multi-task policies into a vision-based policy. With BiDexHD, scalable learning of numerous bimanual dexterous skills from auto-constructed tasks becomes feasible, offering promising advances toward universal bimanual dexterous manipulation. Our empirical evaluation of the TACO dataset, spanning 141 tasks across six categories, demonstrates a task fulfillment rate of **74.59%** on trained tasks and **51.07%** on unseen tasks, showcasing the effectiveness and competitive zero-shot generalization capabilities of BiDexHD. For videos and more information, visit our project page.

## 1 Introduction

Bimanual manipulation is crucial and beneficial. Humans use both hands to do manipulations like using scissors, tying shoelaces, or operating kitchen utensils. The ability to manipulate objects with two hands is fundamental for everyday tasks, because with both hands, we can not only do some "symmetry" collaborative tasks like carrying a heavy box with two hands, but also do "asymmetry" tasks (Liu et al., 2024a) like twisting a bottle cap, which means one hand acts as an auxiliary hand for stabilizing objects and the other acts as an operator.

With the rapid development of embodied artificial intelligence, robotic bimanual dexterous manipulation is getting more and more important in manufacturing, healthcare, agriculture, construction, and tertiary industry (Zhang et al., 2024b). This emphasizes the effective use of tools or manipulations over objects that are deformable or of irregular shapes, overcoming the limitations of low-DOF end-effectors like grippers. Moreover, it addresses complicated human-like hand-object interaction and collaboration. Despite its significance, achieving proficient bimanual manipulation remains a substantial challenge because it severely struggles with the high-dimensional action space. While a line of previous work (Grannen et al., 2023; Yu et al., 2024; Kataoka et al., 2022; Liu et al., 2024a) primarily focuses on bimanual manipulation with grippers, there is still much left to explore for bimanual manipulation with dexterous hands. Previous attempts to solve bimanual dexterous manipulation tasks are mainly based on reinforcement learning (RL) (Lin et al., 2024; Huang et al., 2023; Zhang et al., 2024a). However, they require intricate reward designs tailored to specific manually-designed tasks. Therefore, these approaches lack scalability and generalizability to a broader range of tasks. Recent research (Sindhupathiraja et al., 2024; Fu et al., 2024; He et al., 2024) has advanced robotic bimanual dexterous manipulation through teleoperation. Nevertheless, human intervention is inevitable. We would ask a question:

*"Can we learn diverse bimanual dexterous manipulation skills in a unified and scalable way?"*

Our solution is to use human demonstrations. Compared to robotic demonstrations, human demonstrations are relatively easier to obtain with haptic gloves or MoCap devices rather than deploying a trained policy, and contain much more physically compliant and human-aligned behavior. In this paper, we propose a novel approach to learn diverse bimanual dexterous manipulation skills from human demonstrations. Upon this setting, we propose **BiDexHD**, a unified and scalable framework to automatically turn a human bimanual manipulation dataset into a series of tasks in the simulation and conduct effective policy learning. The majority of previous bimanual studies primarily focus on existing benchmarks or a limited range of tasks. For RL-based methods (Lin et al., 2024; Huang et al., 2023; Zhang et al., 2024a), they tailor specific reward function to specific tasks. For IL-based methods (Wang et al., 2024a; Cheng et al., 2024), it is inevitable to collect a bulk of data for learning specific tasks (typically around 50 trajectories for each single task). In contrast, BiDexHD does not depend on manually-designed tasks or pre-defined tasks in existing benchmarks and instead consistently constructs feasible tasks from any bimanual manipulation trajectory. Furthermore, BiDexHD gets rid of task-specific reward engineering and instead utilize a unified reward function for all automatically constructed bimanual tasks. In a word, BiDexHD breaks the bottleneck of limited tasks and label-intensive manual designs, which is significant to the further development of general-purpose bimanual dexterous manipulation. Though promising, several challenges must be addressed to fully realize this. It is essential to figure out how to accurately mimic fine-grained bimanual behaviors from human demonstrations and avoid collisions and disturbances while encouraging smooth trajectories and synchronous collaboration between both hands. To address this, we carefully design a general two-stage reward function to assign curricula for RL training.

To sum up, our key contributions can be summarized as follows:

- We formalize the problem of learning bimanual dexterous skills from human demonstrations as a preliminary attempt towards universal bimanual skills.

- We propose **BiDexHD**, a unified and scalable reinforcement learning framework for learning diverse bimanual dexterous manipulation from human demonstrations, advancing the capabilities of robots in performing bimanual cooperative tasks.

- We evaluate BiDexHD across 141 auto-constructed tasks over 6 categories from the TACO (Liu et al., 2024b) dataset and demonstrate the superior training performance and competitive generalization capabilities of BiDexHD.

## 2 RELATED WORK

### 2.1 BIMANUAL DEXTEROUS MANIPULATION

In recent years, the robotics community has increasingly focused on dexterous manipulation due to its remarkable flexibility and human-like dexterity. Researchers have developed methods using dexterous hands for tasks such as in-hand manipulation (Arunachalam et al., 2023; Yin et al., 2023; Handa et al., 2023; Qi et al., 2023; Chen et al., 2023; 2022), grasping (Xu et al., 2023; Wan et al., 2023; Qin et al., 2023a; Ye et al., 2023; Qin et al., 2022a), and manipulating deformable objects (Bai et al., 2016; Ficuciello et al., 2018; Li et al., 2023; Hou et al., 2019). However, most existing work focuses on a single dexterous hand, revealing the potential of bimanual dexterity. In fact, for humans, bimanual collaboration takes place frequently in daily life such as riding, carrying heavy objects, and using tools. There are heterogeneous research directions towards bimanual dexterous manipulation. Some researchers attempt to settle down to specific tasks via reinforcement learning. For example, recent work (Lin et al., 2024) investigates twisting lids with two multi-fingered hands, DynamicHandover (Huang et al., 2023) explores throwing and catching, and ArtiGrasp (Zhang et al., 2024a) focuses on a few grasping and articulation tasks. Gbagbe et al. (2024) leveraged large language models to design a system for bimanual robotic dexterous manipulation, while Wang et al. (2024b) proposed to solve bimanual grasping via physics-aware iterative learning and prediction of saliency maps. A recent work (Gao et al., 2024) adopts keypoints-based visual imitation learning to learn bimanual coordination strategies. Unlike existing work, in this paper, we offer a general solution to learn from bimanual demonstrations by designing a unified reward function to learn a state-based policy via reinforcement learning and distilling it into a vision-based policy.

## 2.2 Learning Dexterity From Human Demonstrations

As a sample-efficient data-driven way, learning from human demonstrations has been proven successful in robot learning (Jia et al., 2024; Mandlekar et al., 2023; Odesanmi et al., 2023). Compared with learning dexterity via reinforcement learning which is notoriously challenging for policy learning due to the high degrees of freedom and the necessity of manually designing task-specific reward functions, learning complex dexterous behaviors from diverse accessible human demonstrations (Smith et al., 2019; Schmeckpeper et al., 2020; Shao et al., 2021) is a more stable and scalable approach. A line of previous studies (Arunachalam et al., 2023; Mandikal & Grauman, 2021; Sivakumar et al., 2022; Qin et al., 2022b; Mandikal & Grauman, 2022; Liu et al., 2023; Shaw et al., 2023b; Chen et al., 2024) explicitly leverages human demonstrations to facilitate the acquisition of dexterous manipulation skills mainly by human-robot-arm-hand retargeting and imitation learning. However, these studies predominantly focus on single-hand manipulation and are often limited to tasks such as in-hand manipulation (Arunachalam et al., 2023) or video-conditioned teleoperation (Sivakumar et al., 2022). With the recent advent of diverse and comprehensive human bimanual manipulation datasets (Zhan et al., 2024; Liu et al., 2024b; Fan et al., 2023; Razali & Demiris, 2023) which naturally provide a rich resource for high-quality posture sequences of dual hands and bimanual interaction with diverse real objects, a lot of bimanual manipulation tasks can be automatically defined. Thus, in this work, we aim to address more challenging and general bimanual dexterous skill learning purely based on automatically constructed tasks from human demonstrations.

## 3 Preliminaries

### 3.1 Task Formulation

**Dec-POMDP**. We formulate each bimanual manipulation task as a decentralized partially observable Markov decision process (Dec-POMDP). The Dec-POMDP can be represented by the tuple $Z = (\mathcal{N}, \mathcal{M}, S, \boldsymbol{O}, \boldsymbol{A}, P, R, \rho, \gamma)$. Dual hands with arms are separated as $\mathcal{N}$ agents, which is represented by set $\mathcal{M}$. The proprioception of robots and the information about objects are initialized at $s_0 \in S$ according to the initial state distribution $\rho(s_0)$. At each time step $t$, the state is represented by $s_t$, and the $i$-th agent receives an observation $o_t^i \in \boldsymbol{O}$ based on $s_t$. Subsequently, the policy of the $i$-th agent, $\pi_i \in \boldsymbol{\Pi}$, takes $o_t^i$ as input and outputs an action $a_t^i \in A^i$. The joint action of all agents is denoted by $\boldsymbol{a}_t \in \boldsymbol{A}$, where $\boldsymbol{A} = A^1 \times A^2 \times \cdots A^{\mathcal{N}}$. The state transits to the next state according to the transition function $s_{t+1} \sim P(s_{t+1}|s_t, \boldsymbol{a}_t)$. After this, the $i$-th agent receives a reward $r_t^i$ based on the reward function $R(s_t, \boldsymbol{a}_t)$. The objective is to find the optimal policy $\boldsymbol{\pi}$ that maximizes the expected sum of rewards $\mathbb{E}_{\boldsymbol{\pi}}[\sum_{t=0}^{T-1} \gamma^t \sum_{i=1}^{\mathcal{N}} r_t^i]$ over an episode with $T$ time steps, where $\gamma$ is the discount factor.

**Environment Setups**. The leftmost subgraph in Figure 2 illustrates the setups for each bimanual manipulation task in IsaacGym (Makoviychuk et al., 2021). In general, there are a tool and a target object initialized on a table. $\mathcal{N} = 2$ robotic arms are installed in front of the table, with left LEAP Hand (Shaw et al., 2023a) mounted on the left arm and right hand on the right arm. The right hand reaches for the tool, and the left hand targets the object. Both hands coordinate to simultaneously move, pick up, and manipulate the objects above the table. Note that our method applies to all dexterous hand embodiments. The observation space $\boldsymbol{O}$ contains robot proprioception and object information. The left and right policies both output 22 joint angles normalized to $[-1, 1]$, and the robots are controlled via position control. See more details in Appendix B.4.

**Dataset Preparation**. A human bimanual manipulation dataset consists of $M$ trajectories $\mathcal{D} = \{\tau^1, \tau^2, \ldots, \tau^M\}$, each of which describes a human using a tool with his right hand to manipulate a target object with his left hand. The behavior of each trajectory can be recapped with a triplet (action, tool, object). Any triplet belongs to a union $\mathcal{U} = \mathcal{V} \times \Omega \times \Omega$, where $\Omega$ denotes the set of all objects and tools, and $\mathcal{V}$ denotes the set of all human actions. According to different behaviors depicted in $\mathcal{V}$, we can split all the tasks into $|\mathcal{V}|$ categories. Each trajectory $\tau^i = \{\mathbf{h}^{\text{tool}}, \mathbf{h}^{\text{object}}, \hat{\mathbf{x}}_t^{\text{tool}}, \hat{\mathbf{q}}_t^{\text{tool}}, \hat{\mathbf{x}}_t^{\text{object}}, \hat{\mathbf{q}}_t^{\text{object}}, \Theta_t^{\text{left}}, \Theta_t^{\text{right}}\}_{t:1..N}^i$ involves a pair of meshes of the tool and object from a object mesh set $\mathbf{h}^{\text{tool}}, \mathbf{h}^{\text{object}} \in \mathcal{H}$, $N$-step position $\mathbf{x} \in \mathbb{R}^3$ and orientation $\mathbf{q} \in \mathbb{R}^4$ sequence of the tool and the object, and the pose sequence of hands described in MANO (Romero et al., 2017) parameters $\Theta$.

**Figure 1:** The three-phase framework, BiDexHD, unifies constructing and solving tasks from human bimanual datasets instead of existing benchmarks. In phase one, BiDexHD constructs each bimanual task from a human demonstration. In phase two, BiDexHD learns diverse state-based policies from a generally designed two-stage reward function via multi-task reinforcement learning. A group of learned policies are then distilled into a vision-based policy for inference in phase three.

## 3.2 TEACHER-STUDENT LEARNING

It is well known (Chen et al., 2022; 2023) that directly learning a multi-task vision-based policy for dexterous hands is extremely challenging. A more popular and scalable approach is teacher-student learning (Wan et al., 2023), which not only simplifies the complexity of multi-DoF robot multi-task learning but also enhances the efficiency of point cloud-based policy learning. In the teacher learning phase, a single state-based policy is first trained via reinforcement learning, leveraging privileged information to solve multiple similar tasks. Once trained, multiple teacher policies can effectively tackle all tasks. In the student learning phase, a vision-based student policy is distilled from the teacher policies. A key distinction between teacher and student observations is how object information is represented. While the teacher's observation space includes precise details about an object's position, orientation, and linear and angular velocities, the student's observation relies on point clouds consisting of $P$ sampled points from the object's surface mesh. In this way, the learned student policy is promising to be deployed in real world to deal with multiple tasks provided that the real point clouds can be constructed from real-time multi-view RGB-D camera system.

## 4 LEARNING BIMANUAL DEXTERITY FROM HUMAN DEMONSTRATIONS

### 4.1 OVERVIEW

As illustrated in Figure 1, we propose a scalable three-phase framework. In the first phase, we parallelize the construction of Dec-POMDP bimanual tasks from a human bimanual manipulation dataset within IsaacGym (Makoviychuk et al., 2021). After task initialization, the subsequent two phases adopt a teacher-student policy learning framework. Following the approach of Chen et al. (2022; 2023); Wan et al. (2023), we utilize Independent Proximal Policy Optimization (IPPO) (De Witt et al., 2020) during the second phase to independently train state-based teacher policies for constructed bimanual dexterous tasks in parallel. Each expert focuses on a subset of tasks that require similar behaviors. In the final phase, the teacher policies are distilled into a vision-based student policy, integrating skills across related tasks.

### 4.2 TASK CONSTRUCTION FROM BIMANUAL DATASET

In this work, we primarily focus on bimanual tool using tasks. Recent datasets (Liu et al., 2024b; Zhan et al., 2024; Fan et al., 2023; Razali & Demiris, 2023) capture a wide range of bimanual cooperative behaviors, involving the use of tools to manipulate objects via motion capture and 3D scanning. The rich data, including object pose trajectories and hand-object interaction postures, provides sufficient information to construct feasible bimanual tasks. The task construction process from the bimanual dataset involves data preprocessing and simulation initialization.

**Data Preprocessing**. We extract the wrist and fingertip pose of dual hands at each timestep $\{V_t^{\mathrm{side}}, J_t^{\mathrm{side}}\} = \mathrm{MANO}(\Theta_t^{\mathrm{side}})$, side $\in \{\mathrm{left}, \mathrm{right}\}$ with MANO (Romero et al., 2017) parameters $\Theta = \{\alpha, \beta, \hat{\mathbf{x}}^{\mathrm{w}}\}$, where $\alpha \in \mathbb{R}^{48}, \beta \in \mathbb{R}^{10}$, and $\hat{\mathbf{x}}^{\mathrm{w}} \in \mathbb{R}^3$ represent hand pose, hand shape parameters, and wrist position respectively. $V \in \mathbb{R}^{778\times 3}$ and $J \in \mathbb{R}^{21\times 3}$ represent vertices and joints on a hand respectively. The quaternion of the wrist $\hat{\mathbf{q}}^{\mathrm{w}} \in \mathbb{R}^4$ is translated from axis-angle $\beta_{0:3}$. Given that single LEAP Hand (Shaw et al., 2023a) has only four fingers, we can easily filter the corresponding positions of these $m = 4$ fingers in $J$, denoting them as $\mathbf{x}^{\mathrm{ft}} \in \mathbb{R}^{m\times 3}$. In the following sections, $\tau^i$ is denoted as:

$$\tau^i = \{\hat{\mathbf{x}}_t^{\mathrm{tool}}, \hat{\mathbf{q}}_t^{\mathrm{tool}}, \hat{\mathbf{x}}_t^{\mathrm{object}}, \hat{\mathbf{q}}_t^{\mathrm{object}}, \hat{\mathbf{x}}_t^{\mathrm{lw}}, \hat{\mathbf{q}}_t^{\mathrm{lw}}, \hat{\mathbf{x}}_t^{\mathrm{rw}}, \hat{\mathbf{q}}_t^{\mathrm{rw}}, \hat{\mathbf{x}}_t^{\mathrm{lft}}, \hat{\mathbf{x}}_t^{\mathrm{rft}}\}_{t:1..N}^i. \qquad (1)$$

**Simulation Initialization**. After data preprocessing, we can construct bimanual manipulation tasks $\Gamma = \{\mathcal{T}^1, ..., \mathcal{T}^M\}$ in Issac Gym in parallel. For each task $\mathcal{T}^i$, the mesh of a tool $\mathbf{h}^{\mathrm{tool}}$ and a target object $\mathbf{h}^{\mathrm{object}}$, along with two arms with hands are initialized with a fixed initial observation vector:

$$o_0^{\mathrm{side}} = \{(\mathbf{j}, \mathbf{v})^{\mathrm{side}}, (\mathbf{x}, \mathbf{q})^{\mathrm{side,w}}, \mathbf{x}^{\mathrm{side,ft}}, (\mathbf{x}, \mathbf{q}, \mathbf{v}, \mathbf{w}, \mathrm{id})^{\mathrm{obj}}\}_0^{\mathrm{side}}$$
$$\text{where side, obj} \in \{(\mathrm{left, object}), (\mathrm{right, tool})\}. \qquad (2)$$

The robot proprioception includes arm-hand joint angles and velocities, wrist poses, and fingertip positions, and object information includes object positions, orientations, linear and angular velocities, and a unique object identifier for multi-task learning. For all tasks, $(\mathbf{j}, \mathbf{v})_0$ are all reset to zero. The initial states of wrist and fingertips are calculated with forward kinematics accordingly. Except identifiers, the initial observations for all tools and target objects keep unchanged. It is worth noting that we assume the robot to be right-handed by default, *i.e.*, the left hand handles the target object and the right hand handles the tool. For brevity, the repeated notation side, obj $\in \{(\mathrm{left, object}),(\mathrm{right, tool})\}$ is omitted in subsequent sections.

To ensure the feasibility of each task, after initialization, we use retargeting optimizers (Qin et al., 2023b) to map human hand motions to robot hand joint angles and solve inverse kinematics (IK) to determine the robot arm joint angles based on the robot's palm base pose. By replaying all object-hand trajectories in the simulator, we can easily identify and remove invalid tasks to build up a complete task set $\Gamma$.

### 4.3 MULTI-TASK STATE-BASED POLICY LEARNING

In the second phase, we focus on learning a multi-task state-based policy for tasks that require similar behaviors. Broadly, these tasks can generally be divided into two stages: first, aligning the simulation state with initial $\tau_0^i$ of a trajectory, and second, following each step of the trajectory. During the alignment stage, both hands should prioritize approaching their objects as quickly as possible. The left hand learns to grasp or stabilize the target object, while the right hand learns to grasp the tool. Once simulation alignment is achieved, both hands are expected to maintain their hold and follow the pre-defined trajectory derived from the human demonstration dataset to perform the manipulations in sync. The pipeline is illustrated in Figure 2. We initialize objects and robots at stage zero, finish simulation alignment at stage one, and conduct trajectory tracking at stage two via IPPO to learn state-based policies $\pi_\theta^{\mathrm{side}}(\mathbf{a}_t^{\mathrm{side}}|o_t^{\mathrm{side}}, \mathbf{a}_{t-1}^{\mathrm{side}})$ conditioning on the current observation $o_t^{\mathrm{side}} = \{(\mathbf{j}, \mathbf{v})^{\mathrm{side}}, (\mathbf{x}, \mathbf{q})^{\mathrm{side,w}}, \mathbf{x}^{\mathrm{side,ft}}, (\mathbf{x}, \mathbf{q}, \mathbf{v}, \mathbf{w}, \mathrm{id})^{\mathrm{obj}}\}_t^{\mathrm{side}}$ and previously executed action $\mathbf{a}_{t-1}^{\mathrm{side}}$ for dual hands.

**Stage 1: Simulation Alignment**. The central goal of stage one is to align the state of simulation to the first step in a trajectory by moving the tool and target object from the fixed initial pose to $\tau_0$, which serves as an essential yet challenging prerequisite for subsequent trajectory tracking in stage two. Through experiments in Section 5.4, we find that it is not feasible to directly acquire dynamic skills from static poses through imitation. Instead, we adopt reinforcement learning to develop skills like grasping, twisting and pushing. Some previous work (Luo et al., 2024; Xu et al., 2023) on grasping prefers introducing additional pre-grasp poses by estimating grasping pose upon manipulated objects. We adopt a simpler but more generalizable approach by learning skills directly from the object poses provided in the dataset. Specifically, we anchor the first timestep in the dataset as the reference timestep to establish a tool-object reference pose pair for each manipulation task. Stage one is considered complete once both the tool and the object reach the specified pose for a sustained $u$-step duration. Rewards are carefully designed to encourage the object to be lifted above

**Figure 2:** General two-stage teacher learning. For each task $\mathcal{T}^i$, at stage zero, all joint poses are initialized to the zero pose. Both the tool and object are initialized to poses sampled from a fixed Gaussian distribution centered at a fixed value with added small noise. At stage one, approaching reward $r_{\text{appro}}$ encourages both hands to get close to their grasping centers $\hat{\mathbf{x}}_{\text{gc}}$, and lifting reward $r_{\text{lift}}$ along with extra bonus $r_{\text{bonus}}$ incentivizes moving both objects to thier reference poses respectively. After simulation alignment, dual hands will manipulate objects under the guidance of tracking reward $r_{\text{track}}$.

the table in reference to the filtered reference poses. The total reward consists of an approaching reward, a lifting reward, and a bonus reward.

The approaching reward, $r_{\text{appro}}$, encourages both dexterous hands to approach and remain close to the object. In other words, the goal is to minimize the distance between the robot's palm, fingertips, and the grasp center. Since functional grasping is critical for tool using, we do not simply select the geometric center of the object. Instead, we pre-compute the grasping center $\hat{\mathbf{x}}_{\text{gc}}$ for each tool and object based on the dataset. Specifically, for each task, we use the human-demonstrated wrist and fingertip positions at the reference timestep–$\hat{\mathbf{x}}_0^{\text{lw}}, \hat{\mathbf{x}}_0^{\text{rw}}, \hat{\mathbf{x}}_0^{\text{lft}}, \hat{\mathbf{x}}_0^{\text{rft}}$–as anchor points. We then uniformly sample 1024 points from the surface of the object mesh $\mathbf{h}^{\text{tool}}, \mathbf{h}^{\text{object}}$ to form a representative point set $\mathcal{P}$ and compute the average grasp center based on the top $L = 50$ nearest points. $r_{\text{appro}}$ penalizes the distance between the wrist, fingertips, and the grasp center, and is defined as

$$r_{\text{appro}}^{\text{side}} = -\|\mathbf{x}_t^{\text{side,w}} - \hat{\mathbf{x}}_{\text{gc}}^{\text{obj}}\|_2 - w_r \sum^m \|\mathbf{x}_t^{\text{side,ft}} - \hat{\mathbf{x}}_{\text{gc}}^{\text{obj}}\|_2$$
$$\text{where } \hat{\mathbf{x}}_{\text{gc}}^{\text{obj}} = \frac{1}{L} \sum \text{NN}\left(\mathcal{P}, L, \frac{\hat{\mathbf{x}}_0^{\text{side,w}} + \sum^m \hat{\mathbf{x}}_0^{\text{side,ft}}}{m+1}\right). \quad (3)$$

The lifting reward $r_{\text{lift}}$ encourages holding objects tightly in hands and lifting to desired reference poses. As long as the lifting conditions are satisfied, the robots receive a lifting reward $r_{\text{lift}}$ composed of a non-negative linear position reward and a negative quaternion distance reward,

$$r_{\text{lift}}^{\text{side}} = \begin{cases} r_{\text{pos}}^{\text{side}} + w_q r_{\text{quat}}^{\text{side}} & \text{if } \mathbb{I}\left(\|\mathbf{x}_t^{\text{side,w}} - \hat{\mathbf{x}}_{\text{gc}}^{\text{obj}}\|_2 \leq \lambda_{\text{w}} \cap \sum^m \|\mathbf{x}_t^{\text{side,ft}} - \hat{\mathbf{x}}_{\text{gc}}^{\text{obj}}\|_2 \leq \lambda_{\text{ft}}\right) \\ 0 & \text{otherwise} \end{cases}$$
$$\text{where } r_{\text{pos}}^{\text{side}} = \max\left(1 - \frac{\|\mathbf{x}_t^{\text{obj}} - \hat{\mathbf{x}}_0^{\text{obj}}\|_2}{\|\mathbf{x}_0^{\text{obj}} - \hat{\mathbf{x}}_0^{\text{obj}}\|_2}, 0\right), \quad r_{\text{quat}}^{\text{side}} = -\mathbb{D}_{\text{quat}}\left(\mathbf{q}_t^{\text{obj}}, \hat{\mathbf{q}}_0^{\text{obj}}\right). \quad (4)$$

Here, $\mathbf{x}_0^{\text{object}}$ and $\mathbf{x}_0^{\text{tool}}$ respectively represent the initial positions of the target object and tool in the simulator, while $\hat{\mathbf{x}}_0$ denotes the first reference position in a human demonstration.

The bonus reward $r_{\text{bonus}}$ incentivizes the target object or the tool to reach and finally stay at their reference poses, lays a foundation for the second manipulation stage. $r_{\text{bonus}}$ becomes positive only when the distance between an object's current position and its reference position becomes lower than $\varepsilon_{\text{succ}}$. Stage one is considered successful only if both $r_{\text{bonus}}^{\text{left}}$ and $r_{\text{bonus}}^{\text{right}}$ are positive for at least $u$ consecutive steps. Thus, the bonus reward $r_{\text{bonus}}$ is defined as

$$r_{\text{bonus}}^{\text{side}} = \begin{cases} \frac{1}{1 + \|\mathbf{x}_t^{\text{obj}} - \hat{\mathbf{x}}_0^{\text{obj}}\|_2} & \text{if } \mathbb{I}\left(\|\mathbf{x}_t^{\text{obj}} - \hat{\mathbf{x}}_0^{\text{obj}}\|_2 \leq \varepsilon_{\text{succ}}\right) \\ 0 & \text{otherwise.} \end{cases} \quad (5)$$

The total alignment reward is the linear weighted sum of the three components.

$$r_{\text{align}}^{\text{side}} = w_1 r_{\text{appro}}^{\text{side}} + w_2 r_{\text{lift}}^{\text{side}} + w_3 r_{\text{bonus}}^{\text{side}} \quad (6)$$

**Stage 2: Trajectory Tracking**. Once stage one is completed, the left hand is securely holding the target object, and the right hand keeps grasping the tool at its desired reference pose. The next step is to maintain the grasp and follow a trajectory to perform the manipulation. To achieve this, we design a more fine-grained exponential reward, $r_{\text{track}}$, which encourages the dexterous hands to precisely track the desired positions at each timestep in a trajectory starting from the reference timestep. Assuming that human hands are more flexible than robotic hands, we introduce a constant tracking frequency $f$, where $f$ simulation steps correspond to one step in the dataset. Let $\hat{\mathbf{x}}_i^{\text{obj}}$ represent the position of a object at $i$-th step in a $l$-step human-demonstrated trajectory and $\mathbf{x}_{t_i}^{\text{obj}}$ represent the object's position at the corresponding simulation step in IsaacGym. We have $i = \lceil t_i/f \rceil \in [0, l)$, and the tracking reward is defined as

$$r_{\text{track}}^{\text{side}} = \begin{cases} \exp\left(-w_t \|\mathbf{x}_{t_i}^{\text{obj}} - \hat{\mathbf{x}}_i^{\text{obj}}\|_2\right) & \text{if stage 1 succeeds} \\ 0 & \text{otherwise.} \end{cases} \tag{7}$$

We adopt IPPO to learn a unified policy from the combination of all rewards for the two stages,

$$r_{\text{total}}^{\text{side}} = r_{\text{align}}^{\text{side}} + w_4 r_{\text{track}}^{\text{side}}. \tag{8}$$

$r_{\text{total}}$ unifies two stages of bimanual dexterous manipulation, enabling scaling up to multi-task policy learning for a wide range of constructed bimanual tasks.

### 4.4 VISION-BASED POLICY DISTILLATION

We employ DAgger (Ross et al., 2011), an on-policy imitation learning algorithm, to develop a vision-based policy for each task category $\nu \in \mathcal{V}$, under the supervision of a group of state-based teacher policies. To enhance generalization capabilities for new objects or unseen tasks, we propose transforming the student policy into a trajectory-conditioned in-context policy, denoted as $\pi_\phi^{\text{side}}(\mathbf{a}_t^{\text{side}}|\mathbf{o}_t^{\text{side}}, \mathbf{p}_t^{\text{side}}, \mathbf{a}_{t-1}^{\text{side}})$, where $\mathbf{o}_t = \{(\mathbf{j}, \mathbf{v})^{\text{side}}, (\mathbf{x}, \mathbf{q})^{\text{side,w}}, \mathbf{x}^{\text{side,ft}}, \text{pc}^{\text{obj}}\}_t$, $K$-step future pose $\mathbf{p}_t^{\text{side}} \in \mathbb{R}^{K \times 3}$, and $\text{pc}_t^{\text{obj}} \in \mathbb{R}^{P \times 3}$. Specifically, to get point clouds $\text{pc}_t^{\text{tool}}$ and $\text{pc}_t^{\text{object}}$, we pre-sample 4096 points from the surface of $\mathbf{h}^{\text{tool}}$ and $\mathbf{h}^{\text{object}}$ for each task during initialization. At each timestep $t$, a subset of points are sampled from the pre-sampled point clouds, transformed according to current object pose and added with Gaussian noise for robustness. Besides, it is important to note that during DAgger distillation, we augment traditional vision-based policy $\pi_\phi^{\text{side}}(\mathbf{a}_t^{\text{side}}|\mathbf{o}_t^{\text{side}}, \mathbf{a}_{t-1}^{\text{side}})$ with next $K$ positions along the object's trajectory as additional inputs. This design allows the learned policy to utilize more information about the motion of objects, such as movement direction and speed in the near future, facilitating zero-shot transfer to unfamiliar tasks or objects. Notably, we can easily mask this additional input by setting $K = 0$. We further investigate the influence of $K$ future steps in Section 5.4. The whole teacher-student training process is summarized in Appendix A. More implementation details can be found in Appendix B.

## 5 EXPERIMENTS

### 5.1 SETUPS

**Dataset**. We evaluate the effectiveness of BiDexHD on the TACO (Liu et al., 2024b) dataset, a large-scale bimanual manipulation dataset that encompasses diverse human demonstrations using tools to manipulate target objects in real-world scenarios. BiDextHD converts 6 categories $\mathcal{V} = \{\text{Dust, Empty, Pour in some, Put out, Skim off, Smear}\}$ of total 141 human demonstrations in the TACO dataset to Dec-POMDP tasks (See Appendix D for task examples). Task diversity and abundance make BiDexHD easy to scale up. All tasks can be separated into 16 semantic groups, each of which gathers a number of similar demonstrations with the same action, the same tool-object category but different tool and object instances. BiDextHD constructs a task from single demonstration, and thus each semantic group correspond to a semantic subtask. We adopt teacher-student learning to train 16 semantic sub-tasks and distill teacher policies with similar skills into 6 vision-based policies for each category eventually.

To evaluate the effectiveness of the framework as well as the generalizability of the learned policies, we split 80% tasks for training (**Train**) and the rest 20% unseen tasks for testing. Detailed descriptions of dataset split are provided in Appendix B.2. For each task in the testing set, if the object and

tool both occur in the training set it is labeled as a kind of combinational task (**Test Comb**), and otherwise it is labeled as a new task (**Test New**).

**Metrics**. To measure the quality of our constructed tasks, we introduce two metrics $r_1$ and $r_2$.

- The first is the average success rate $r_1$ of stage one. For a number of $n$ episodes, $r_1 = \frac{1}{n} \sum_{e=1}^{n} \mathbb{I}_1^e$ averages over the number of episodes that satisfys conditions $\mathbb{I}_1$ at stage one.

$$\mathbb{I}_1 : \quad \exists 0 < t < T-u \quad \sum_t^{t+u} \mathbb{I}\left( \|\mathbf{x}_t^{\text{object}} - \hat{\mathbf{x}}_0^{\text{object}}\|_2 \leq \varepsilon_{\text{succ}} \cap \|\mathbf{x}_t^{\text{tool}} - \hat{\mathbf{x}}_0^{\text{tool}}\|_2 \leq \varepsilon_{\text{succ}} \right) = u$$

- The second is the average tracking rate $r_2$ of stage two. Each task corresponds to $l$-step human-demonstrated trajectory. For each episode, calculate the proportion of steps where two objects both effectively follows their desired poses. $r_2$ is the average tracking rate over $n$ episodes.

$$r_2 = \frac{1}{nl} \sum_{}^{n} \sum_{i=0}^{l-1} \mathbb{I}\left( \|\mathbf{x}_{t_i}^{\text{object}} - \mathbf{x}_i^{\text{object}}\|_2 \leq \varepsilon_{\text{track}} \cap \|\mathbf{x}_{t_i}^{\text{tool}} - \mathbf{x}_i^{\text{tool}}\|_2 \leq \varepsilon_{\text{track}} \right)$$

It is important to note that $r_2$ serves as the primary metric for indicating task completion while $r_1$ is an intermediate metric for assessing task progression. Considering the choice of $\varepsilon_{\text{succ}}$ and $\varepsilon_{\text{track}}$ has a non-legligible impact for the reported results, we will discuss the sensitivity of these thresholds in Section 5.4. By default, we choose $\varepsilon_{\text{succ}} = \varepsilon_{\text{track}} = 0.1$ for evaluation.

## 5.2 TEACHER LEARNING

Upon the framework of BiDexHD, different base RL algorithms can be incorporated. We mainly compare the performance of independent PPO (**BiDexHD-IPPO**) and centralized PPO (**BiDexHD-PPO**). For BiDexHD-IPPO, two agents possess their own observations and execute their own actions. For BiDexHD-PPO, a single policy takes as input both observations and is trained to output all actions that maximize the sum of all total rewards in an episode, which essentially transforms a Dec-POMDP task into a POMDP task.

**Table 1:** The average success rate of stage 1 and tracking rate of stage 2 during training and evaluation across all tasks constructed from the TACO dataset under $\varepsilon_{\text{succ}} = \varepsilon_{\text{track}} = 0.1$.

| Method | Train $r_1(\%)$ | Train $r_2(\%)$ | Test Comb $r_1(\%)$ | Test Comb $r_2(\%)$ | Test New $r_1(\%)$ | Test New $r_2(\%)$ |
|---|---|---|---|---|---|---|
| BiDexHD-PPO | 90.55 | 53.88 | 78.74 | 36.99 | **81.42** | **26.24** |
| BiDexHD-IPPO (w/o stage-1) | 25.00 | 17.52 | 24.80 | 18.10 | 19.85 | 08.51 |
| BiDexHD-IPPO (w/o gc) | 90.53 | 66.39 | 91.47 | 52.11 | 77.03 | 22.63 |
| BiDexHD-IPPO (w/o bonus) | 97.67 | 66.65 | 98.01 | 59.76 | 77.96 | 17.52 |
| **BiDexHD-IPPO** | **98.71** | **78.18** | **98.37** | **59.94** | 75.48 | 21.34 |
| BC | 00.00 | 00.00 | 00.00 | 00.00 | 00.00 | 00.00 |
| BiDexHD-PPO+DAgger | 95.35 | 55.82 | 76.75 | 30.42 | 86.34 | 30.00 |
| **BiDexHD-IPPO+DAgger** | **99.38** | **74.59** | **92.85** | **48.43** | **94.79** | **53.71** |

**RL Results**. The first and last rows in the green section of Table 1 present the average performance across all auto-constructed bimanual tasks. For tasks with seen objects (Train and Test Comb), BiDexHD-IPPO nearly completes stage 1 by successfully reaching the reference poses and maintaining high-quality tracking during stage 2, which demonstrates its impressive scalability across diverse tasks in the TACO dataset. In contrast, BiDexHD-PPO underperforms compared to BiDexHD-IPPO, particularly on tasks with seen objects. This discrepancy arises because BiDexHD-IPPO is more efficient at acquiring robust skills within limited updates by independently learning left and right policies across a wide range of tasks with smaller observation and action spaces. Furthermore, two independent expert policies focusing solely on specific groups of target objects or tools adapt more easily to similar combinational tasks than a single policy that must attend to both. Consequently, we select IPPO as our base RL algorithm. Detailed evaluation results are recorded in

Appendix C.1. We observe that particularly in 'Pour in some' tasks, the task diversity is relatively small. Efficient BiDexHD-IPPO can achieve overwhelming advantages over BiDexHD-PPO.

When applied to tasks with new objects (Test New), both BiDexHD-IPPO and BiDexHD-PPO experience a noticeable performance decline. The primary reason for this drop is that these approaches incorporate one-hot object labels in observations during state-based training, leading the policy to heavily rely on this information. As a result, during evaluation, the introduction of new labels disrupts decision-making. Therefore, we remove one-hot object labels during policy distillation to enhance generalization.

## 5.3 Ablations on Teacher Learning

We conduct ablation studies focusing on the key designs at stage one during teacher learning.

**Alignment Stage**. To demonstrate the necessity of the design of dataset-simulation alignment stage, we compare BiDexHD-IPPO with a more naive version, denoted as **(w/o stage-1)**, which retains only $r_{\text{track}}$ in RL training at stage 2 and maintains a fixed number of free exploration steps at stage 1. The second line in the green section of Table 1 reveals a significant performance decline. We observe that only 30.5% of relatively easy tasks (See Appendix C.1 for details) achieve positive $r_1$ and $r_2$, while for the remaining tasks, the success rate of stage 1 and the tracking rate of stage 2 remain at zero. This emphasizes the importance of $r_{\text{align}}$ during stage 1.

**Functional Grasping Center**. In BiDexHD, we pre-compute the grasping center $\hat{\mathbf{x}}_{\text{gc}}$ to calculate $r_{\text{appro}}$ in Equation 3. In this section, we explore replacing the grasping center with the object geometric center, denoted as **(w/o gc)**. The results presented in the third line of Table 1 show a decrease in $r_1$ and $r_2$, particularly on tasks involving seen objects compared to BiDexHD-IPPO. To further investigate their discrepancy in behavior, we visualize their grasping poses for a typical task (dust, brush, pan) in Figure 3. BiDexHD-IPPO tends to align more closely with the calculated grasping centers (red points), exhibiting human-like grasping behavior. In contrast, BiDexHD-IPPO (w/o gc) with geometric centers (green points) struggles to find proper poses for using the brush or holding the pan. In fact, the geometric center of an object does not often fall within areas suitable for manipulation. These findings highlight the significance of incorporating a functional grasping center, particularly for objects that are thin, flat, or equipped with handles.

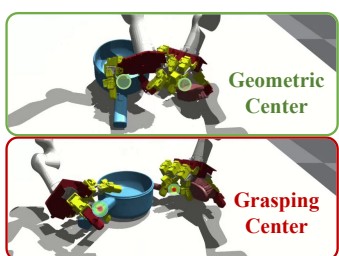

**Figure 3:** A comparison of grasping pose during policy deployment between BiDexHD-IPPO (w/o gc) and BiDexHD-IPPO.

**Success Bonus**. The 4th line in the green section of Table 1 investigates whether removing reward $r_{\text{bonus}}$ defined in Equation 5 affects performance. We observe a decline in $r_2$ on both the training set and unseen tasks involving new objects. We analyze the additional bonus in Equation 5 effectively signals the transition between the two stages, enhancing the policy's awareness of task progression.

## 5.4 Student Learning

For the BiDexHD variants, several trained multi-task state-based teacher policies from one task category are distilled into a single vision-based policy, which is then tested on all tasks. We also introduce behavior cloning (BC) as our baseline. To directly learn bimanual skills from a dataset, we employ Dexpilot (Handa et al., 2020) to retarget human hand motions in the TACO dataset to joint angles for dexterous hands, solving inverse kinematics (IK) for arm joint angles. All joint angles are collected and replayed in IsaacGym (Makoviychuk et al., 2021) to gather observations. BC learns purely from this static observation-action dataset and is ultimately tested under the same configuration as BiDexHD.

**DAgger Results**. The blue section of Table 1 displays the performance of the vision-based policies. Our **BiDexHD-IPPO+DAgger** significantly outperforms both PPO variant and BC, achieving a high task completion rate on the training set and an average $r_2 = 51.07\%$ across all unseen tasks (Test Comb and Test New). This evidence indicates the scalability and competitive generalization ability of BiDexHD framework. Among unseen tasks, we observe a slight decline in $r_2$ for combinational

tasks, while tasks involving new objects show a sharp increase in $r_2$. This suggests that the vision-based policy relies more on information from the point clouds, such as shape and local features, rather than specific one-hot identifiers, enabling effective zero-shot generalization. Conversely, BC performs poorly due to the loss of true dynamics in the simulation, often getting confused by unfamiliar observations and stuck in stationary states. This also reflects the challenges associated with our constructed bimanual tasks. Our framework unifies bimanual skill learning through a combination of trial-and-error and distillation, providing a robust and scalable solution to diverse challenging bimanual manipulation tasks. Detailed evaluation results are reported in Appendix C.2. We observe that in 'Dust' and 'Empty' task categories, the task diversity is relatively ample. Therefore, the distilled policy can surpass the average level of expert policies, which proves that distilling similarity policies bring about positive promotion effects to the final unified policy.

**Table 2:** The metrics of different $K$ future steps under $\varepsilon_{\text{succ}} = \varepsilon_{\text{track}} = 0.1$.

| Metrics (%) | $K$ | | | |
| --- | --- | --- | --- | --- |
| | 0 | 1 | 2 | 5 |
| Train $r_1$ | 98.01 | 98.81 | 98.71 | **99.38** |
| Train $r_2$ | 72.09 | **75.40** | 75.01 | 74.59 |
| Test Comb $r_1$ | **94.36** | 92.11 | 93.26 | 92.85 |
| Test Comb $r_2$ | 46.64 | **49.02** | 48.60 | 48.43 |
| Test New $r_1$ | 93.96 | 94.67 | 94.38 | **94.79** |
| Test New $r_2$ | 49.27 | 51.00 | 50.39 | **53.71** |

**Table 3:** The sensitivity analysis of metrics of BiDexHD-IPPO+DAgger to different $\varepsilon$.

| Metrics (%) | $\varepsilon$ | | |
| --- | --- | --- | --- |
| | 0.05 | 0.075 | 0.1 |
| Train $r_1$ | 96.87 | 98.27 | 99.38 |
| Train $r_2$ | 52.58 | 66.19 | 74.59 |
| Test Comb $r_1$ | 49.30 | 77.74 | 92.85 |
| Test Comb $r_2$ | 13.02 | 24.56 | 48.43 |
| Test New $r_1$ | 79.56 | 88.11 | 94.79 |
| Test New $r_2$ | 17.19 | 37.62 | 53.71 |

**Future Conditioned Steps**. We further examine the selection of $K \in \{0, 1, 2, 5\}$ for future object positions. Specifically, when $K = 0$, the vision-based policy relies exclusively on 3D information from object point clouds and the robot's proprioception. As shown in Table 2, the performance across different values of $K$ does not vary significantly. Even when future conditioned steps are masked ($K = 0$), $r_2$ only exhibits slight declines of 2.5% on trained tasks and an average of 3.1% on all unseen tasks compared to $K = 5$. This evidence suggests that after the multi-task RL training phase, the teachers have acquired diverse and robust skills, making pure imitation sufficient for a student to achieve acceptable performance. Nonetheless, $K$ future steps provide additional informative and fine-grained, albeit implicit, clues such as motion and intention for more precise tracking.

**Discussion**. To investigate the impact of different thresholds on the metrics, we re-evaluate all tasks and report the performance of our BiDexHD-IPPO+DAgger under varying thresholds, $\varepsilon_{\text{succ}} = \varepsilon_{\text{track}} = \varepsilon \in \{0.05, 0.075, 0.1\}$ in Table 3. Notably, stricter metrics have a more pronounced impact on the performance of unseen tasks compared to trained ones, underscoring the challenges of continuous spatial-temporal trajectory tracking in bimanual manipulation tasks. We will focus on addressing more precise behavior tracking in future work.

## 6  CONCLUSION & LIMITATIONS

In this paper, we introduce a novel approach to learning diverse bimanual dexterous manipulation skills that utilizes human demonstrations. Our BiDexHD automatically constructs bimanual manipulation tasks from existing datasets and employs a teacher-student learning approach for a vision-based policy that can tackle similar tasks. Our main technical contributions include designing a unified two-stage reward function for multi-task RL training and an in-context vision-based policy that enhances generalization capabilities. Experimental results demonstrate that BiDexHD facilitates robust RL training and policy distillation, successfully solves six categories of bimanual dexterous manipulation tasks, and effectively transfers to unseen tasks through zero-shot generalization.

Our work forwards a step toward universal bimanual manipulation skills, and some limitations need to be addressed in future research. Exploring strategies for achieving more precise spatial and temporal tracking is a valuable direction for future work. Additionally, incorporating a wider variety of real-world tasks–such as deformable object manipulation and bimanual handover–could reveal further potential in dynamic collaborative manipulation scenarios with bimanual dexterous hands.

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

# A  ALGORITHM

---

**Algorithm 1:** BiDexHD framework.

---

**Input:** Human demonstration dataset $\mathcal{D} = \{\tau^1, \tau^2, \ldots, \tau^M\}$; Object mesh set $\Omega$; State-based policies $\pi_\theta^{\text{side}}$; Vision-based policy $\pi_\phi^{\text{side}}$ (side $\in$ {left,right}).

**Output:** The learned vision-based policy $\pi_\phi^{\text{side}}$.

**Task Construction:**

**for** $\tau^i \in \mathcal{D}$ **do**

    Preprocess each $\tau^i$ by translating MANO parameters to pose sequence in Equation 1 ;

    Construct task $\mathcal{T}^i$ with corresponding $\tau^i$, $\mathbf{h}^{\text{tool}}$ and $\mathbf{h}^{\text{object}}$.

**Teacher learning:**

Sample a subset of similar tasks $\Lambda \subseteq \Gamma = \{\mathcal{T}^1, ..., \mathcal{T}^M\}$;

Parallelly initialize $\mathcal{T}^{i:1...|\Lambda|}$ in IsaacGym simulation with $o_0^{\text{left}}$ and $o_0^{\text{right}}$ in Equation 2;

**while** *not converge* **do**

    Get state-based observations $o_t^{\text{left}}, o_t^{\text{right}}$;

    Sample action $\mathbf{a}_t^{\text{left}} \sim \pi_\theta^{\text{left}}(\mathbf{a}_t^{\text{left}}|o_t^{\text{left}}, \mathbf{a}_{t-1}^{\text{left}}), \mathbf{a}_t^{\text{right}} \sim \pi_\theta^{\text{right}}(\mathbf{a}_t^{\text{right}}|o_t^{\text{right}}, \mathbf{a}_{t-1}^{\text{right}})$;

    Step the environments to observe $o_{t+1}^{\text{left}}, o_{t+1}^{\text{right}}$ and calculate total reward $r_{\text{total}}^{\text{left}}, r_{\text{total}}^{\text{right}}$;

    Save $(o_t^{\text{left}}, \mathbf{a}_t^{\text{left}}, o_{t+1}^{\text{left}}, r_{\text{total}}^{\text{left}})$ and $(o_t^{\text{right}}, \mathbf{a}_t^{\text{right}}, o_{t+1}^{\text{right}}, r_{\text{total}}^{\text{right}})$ into IPPO buffer;

    Update $\pi_\theta^{\text{left}}$ and $\pi_\theta^{\text{right}}$ using IPPO with the IPPO buffer.

**Policy Distillation:**

Index $\{\tau^1, \ldots, \tau^{|\Lambda|}\}$ and pre-sample 4096 points for the tool and target object in each $\mathcal{T}^i$;

Parallelly initialize $\mathcal{T}^{i:1...|\Lambda|}$ in IsaacGym simulation;

**while** *not converge* **do**

    The students get vision-based observations $\mathbf{o}_t^{\text{left}}, \mathbf{o}_t^{\text{right}}$ with sampled point clouds and
    $K$-step future trajectories and sample action

    $\mathbf{a}_t^{\text{left}} \sim \pi_\phi^{\text{left}}(\mathbf{a}_t^{\text{left}}|\mathbf{o}_t^{\text{left}}, \mathbf{a}_{t-1}^{\text{left}}), \mathbf{a}_t^{\text{right}} \sim \pi_\phi^{\text{right}}(\mathbf{a}_t^{\text{right}}|\mathbf{o}_t^{\text{right}}, \mathbf{a}_{t-1}^{\text{right}})$;

    The experts $\pi_\theta^{\text{left}}, \pi_\theta^{\text{right}}$ observe the corresponding $o_t^{\text{left}}, o_t^{\text{right}}$ and labels $\hat{\mathbf{a}}_t^{\text{left}}, \hat{\mathbf{a}}_t^{\text{right}}$;

    Step the environments;

    Save $(\mathbf{o}_t^{\text{left}}, \hat{\mathbf{a}}_t^{\text{left}})$ and $(\mathbf{o}_t^{\text{right}}, \hat{\mathbf{a}}_t^{\text{left}})$ into DAgger buffer;

    Update $\pi_\phi^{\text{right}}$ and $\pi_\phi^{\text{right}}$ by minimizing MSE loss with the DAgger buffer.

---

# B  IMPLEMENTATION DETAILS

## B.1  DATASET PREPROCESSING

**Reference Timestep**. Considering there are a number of useless preparation timesteps before grasping, the reference timestep in Section 4.2 is actually chosen based on the first sudden change of the distance between an object and a tool, because the distance between the tool and object almost stays unchanged before grasping.

**More Details**. We further align the coordinates of human wrist to the coordinates of robot palm base to ensure the same dual-hand manipulation behavior. Besides, due to the geometric discrepancy of objects, we found that the initial height of objects differ a lot in different tasks. Therefore, a translation offset in z-axis is added to all poses in the dataset to keep all the object at the same initial height on the same table.

## B.2  CONSTRUCTED TASKS

**Task Composition**. Table 4 describes the detailed task categories, sub-task names, the split of training and testing set and the diversity of tools and target objects.

**Table 4:** 141 constructed tasks across 6 categories for BiDexHD. "All" refers to the total number of a kind of sub-task. "Train" refers to the number of tasks in the training set. "Test Comb" and "Test New" refer to the number of tasks in two types of testing sets. "Tool" and "Object" refer to number of objects in the corresponding sub-tasks.

| Action | Sub-Task Name | All | Train | Test Comb | Test New | Tool | Object |
|---|---|---|---|---|---|---|---|
| Empty | (empty, bowl, bowl) | 10 | 7 | 1 | 2 | 5 | 5 |
| | (empty, bowl, plate) | 34 | 26 | 1 | 7 | 8 | 9 |
| | (empty, cup, plate) | 1 | 1 | 0 | 0 | 1 | 1 |
| | (empty, teapot, plate) | 12 | 8 | 1 | 3 | 2 | 6 |
| | (empty, teapot, teapot) | 3 | 3 | 0 | 0 | 2 | 2 |
| Pour in some | (pour in some, cup, cup) | 1 | 1 | 0 | 0 | 1 | 1 |
| | (pour in some, cup, plate) | 2 | 2 | 0 | 0 | 2 | 2 |
| | (pour in some, cup, teapot) | 1 | 1 | 0 | 0 | 1 | 1 |
| | (pour in some, teapot, bowl) | 1 | 1 | 0 | 0 | 1 | 1 |
| | (pour in some, teapot, cup) | 2 | 1 | 0 | 1 | 2 | 2 |
| Dust | (dust, brush, bowl) | 20 | 5 | 0 | 15 | 5 | 9 |
| | (dust, brush, pan) | 9 | 6 | 3 | 0 | 4 | 3 |
| Put out | (put out, bowl, bowl) | 10 | 7 | 2 | 1 | 5 | 5 |
| | (put out, bowl, plate) | 16 | 11 | 1 | 4 | 3 | 8 |
| Skim off | (skim off, bowl, plate) | 17 | 12 | 0 | 5 | 5 | 8 |
| Smear | (smear, glue gun, plate) | 2 | 2 | 0 | 0 | 1 | 2 |
| **Total** | – | **141** | 94 | 9 | 38 | – | – |

### B.3 DEXTEROUS HANDS

Currently we use LEAP Hands (Shaw et al., 2023a). In future work, we will introduce more kinds of dexterous hands.

### B.4 SIMULATION SETUP

Two 6-DOF RealMan arms, spaced 0.68 meters apart, are placed in front of a table of 0.7 meters. The 16-DOF LEAP hands are Shaw et al. (2023a) mounted on the left and right arms, with an initial stretching pose. The tool and target object are spaced 0.4m apart horizontally, 0.5m distant from the robotic arm base.

### B.5 TRAINING DETAILS

**BC Details**. To get the arm and hand action labels for imitation learning, we employ Dexpilot (Handa et al., 2020) to retarget human hand motions in the TACO dataset to hand joint angles for dexterous hands and solve inverse kinematics (IK) to convert Mocap 6D wrist pose to 6-DOF arm joint angles. Since each task is built from a single demonstration, we use vanilla imitation learning to directly learn a vision-based policy $\pi_\phi^{side}(\mathbf{a}_t^{side}|\mathbf{o}_t^{side}, \mathbf{a}_{t-1}^{side})$, where the observation $\mathbf{o}_t$ is defined as $\mathbf{o}_t = \{(\mathbf{j}, \mathbf{v})^{side}, (\mathbf{x}, \mathbf{q})^{side,w}, \mathbf{x}^{side,ft}, pc^{obj}\}_t$. The policy is trained for each task from a single observation-action sequence after retargeting. The policy model consists of two 1D convolutional layers to encode point clouds, a dense layer to encode robot states followed by two dense layers to output all joint angles. The architecture (see Appendix B.7), configurations, and hyperparameters are identical to the ones in DAgger vision-based policy learning. The loss function is the standard MSE loss. Experimental results show that imitation learning from a single trajectory fails. To investigate this, we visualize the behavior of the learned BC policy in Figure 4. The results reveal that the learned policy fails to reach or manipulate the object, instead getting stuck in stationary states or self-collision. We identify two primary reasons for this failure. The most obvious one is limited demonstrations. With only one demonstration, large portions of the observation space remain unexplored. As a result, BC struggles with unvisited states. More importantly, lack of kinematics and dynamics affects a lot. Retargeted actions approximate human demonstrations spatially and tempo-

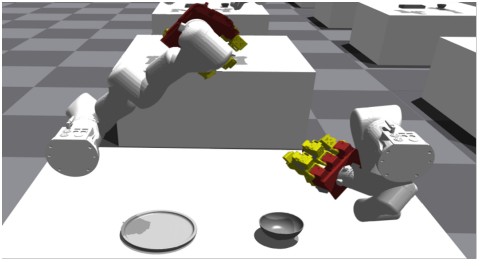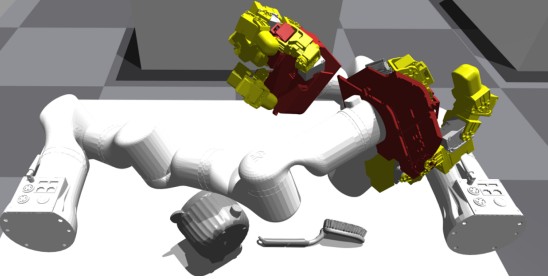

**Figure 4:** Two failure case examples of baseline BC in 'empty' and 'dust' tasks respectively. The learned policy does not show a tendency to reach and manipulate the objects. Instead, the robots tend to get stuck in a stationary state or self-collision.

rally but fail to account for true kinematics and dynamics. This results in fragile policies prone to failure and stationary states, as shown in the video on our project page.

**DAgger Details**. To make student policy learn more efficiently, especially at the early training stage, we mix a few imitation samples into DAgger buffer. Specifically, we choose to use the actions labeled by experts with probability $p = 0.05$ and otherwise the actions that are output by the policy itself. This has often proven desirable in practice, as the naive policy can make more mistakes and visit states that are irrelevant at the early stage of training with relatively few data points (Ross et al., 2011).

**Hyperparameters**. Tables 5 and 6 outline the hyperparameters for IPPO, PPO, DAgger, and BC in BiDexHD, respectively.

**Table 5:** Hyperparameters of IPPO or PPO.

| Hyperparameter | Value |
|---|---|
| $w_r$ | 2.0 |
| $w_t$ | 15.0 |
| $w_1$ | 0.5 |
| $w_2$ | 1.0 |
| $w_3$ | 1.0 |
| $w_4$ | 1.0 |
| $\lambda_w$ | 0.12 |
| $\lambda_{ft}$ | 0.48 |
| Episode Length | 1000 |
| Parallel rollout steps per iteration | 8 |
| Training epochs per iteration | 5 |
| Number of mini-batch | 3 |
| Mini-batch size | 32 |
| Discount factor | 0.96 |
| GAE lambda | 0.95 |
| Clip range | 0.2 |
| Optimizer | AdamW |
| Learning Rate | 3e-4 |
| Number of Environments | 15000 |
| Type of GPUs | A100, or Nvidia RTX 4090 Ti |

## B.6 DATASET EXTENSION

To demonstrate BiDexHD is scalable and transferable in heterogeneous bimanual tasks, we extend our BiDexHD framework to a new bimanual dataset Arctic (Fan et al., 2023), which mainly focuses on bimanual cooperative tasks of a single object. We build up four tasks 'Mixer Holding', 'Capsulemachine Grabbing', 'Box Flipping', 'Ketchup Lifting' from four trajectories in Arctic dataset

**Table 6:** Hyperparameters of DAgger and BC.

| Hyperparameter | Value |
|---|---|
| $P$ | 512 |
| $K$ | 5 |
| Parallel rollout steps per iteration | 8 |
| Training epochs per iteration | 5 |
| Number of mini-batch | 3 |
| Mini-batch size | 32 |
| Optimizer | AdamW |
| Learning Rate | 3e-4 |
| Number of Environments | 5000 |
| Type of GPUs | A100, or Nvidia RTX 4090 Ti |

and follow the pipeline of teacher learning to learn a state-based policy for each task. The average success rate of stage one $r_1$ and trajectory tracking rate $r_2$ shown in Table 7 demonstrate the effectiveness and generalizability of BiDexHD in collaborative bimanual manipulation tasks. Refer to our project page for video demonstrations of Arctic tasks.

**Table 7:** Metrics of BiDexHD-IPPO for four Arctic tasks.

| Task | **Train** $r_1(\%)$ | **Train** $r_2(\%)$ |
|---|---|---|
| Mixer Holding | 90.01 | 79.24 |
| Capsulemachine Grabbing | 96.47 | 93.45 |
| Box Flipping | 94.10 | 91.23 |
| Ketchup Lifting | 93.98 | 82.99 |

### B.7 MODEL ARCHITECTURE

Our codebase for RL and DAgger is built upon UniDexGrasp++ (Wan et al., 2023). For each state-based policy, we employ five-layer multi-layer perceptrons (MLPs) for both the actor and the critic, featuring hidden layers with dimensions [1024, 1024, 512, 512] and using ELU activation functions. For the vision-based policy, we utilize a simplified PointNet (Qi et al., 2017) backbone that incorporates two 1D convolutional layers, a mixture of maximum and average pooling operations, and two MLP layers to process the object point cloud, resulting in an output dimension of 128. Both the actor and the critic share the output of this backbone. The same network architecture is adapted to BC baselines and all PPO variants both for Arctic tasks and TACO tasks.

### B.8 COMPUTATION RESOURCES

we train a state-based IPPO policy for single sub-tasks for around two days, and distill teacher policies into a vision-based policy for each action category for around one day on single 40G A100 GPUs. All the evaluations are done on a 24G Nvidia RTX 4090 Ti GPU for about half an hour.

## C EVALUATION RESULTS

### C.1 RESULTS OF TEACHER LEARNING

Tables below record the detailed evaluation results for each sub-task. '–' represents the absence of testing tasks. The last row of each table shows the average results over all sub-tasks.

**Table 8:** Detailed Metrics of BiDexHD-PPO for each sub-task under $\varepsilon_{\text{succ}} = \varepsilon_{\text{track}} = 0.1$.

| Action | Sub-Task Name | Train $r_1(\%)$ | Train $r_2(\%)$ | Test Comb $r_1(\%)$ | Test Comb $r_2(\%)$ | Test New $r_1(\%)$ | Test New $r_2(\%)$ |
|---|---|---|---|---|---|---|---|
| Empty | (empty, bowl, bowl) | 99.38 | 69.28 | 99.30 | 62.24 | 93.64 | 53.13 |
| | (empty, bowl, plate) | 97.88 | 50.38 | 100.00 | 35.93 | 65.91 | 8.95 |
| | (empty, cup, plate) | 77.03 | 33.41 | – | – | – | – |
| | (empty, teapot, plate) | 100.00 | 82.09 | 71.83 | 23.62 | 96.84 | 28.04 |
| | (empty, teapot, teapot) | 100.00 | 27.89 | 42.42 | 7.63 | – | – |
| Pour in some | (pour in some, cup, cup) | 25.50 | 22.71 | – | – | – | – |
| | (pour in some, cup, plate) | 99.88 | 82.16 | – | – | – | – |
| | (pour in some, cup, teapot) | 100.00 | 69.69 | – | – | – | – |
| | (pour in some, teapot, bowl) | 99.56 | 5.85 | – | – | – | – |
| | (pour in some, teapot, cup) | 85.48 | 31.26 | – | – | – | – |
| Dust | (dust, brush, bowl) | 99.61 | 77.70 | – | – | 78.57 | 17.54 |
| | (dust, brush, pan) | 73.68 | 48.22 | 41.58 | 15.69 | – | – |
| Put out | (put out, bowl, bowl) | 91.54 | 39.80 | 100.00 | 46.25 | 67.74 | 5.48 |
| | (put out, bowl, plate) | 99.57 | 69.75 | 96.08 | 67.54 | 81.51 | 34.51 |
| Skim off | (skim off, bowl, plate) | 99.68 | 74.59 | – | – | 85.76 | 36.01 |
| Smear | (smear, glue gun, plate) | 100.00 | 77.26 | – | – | – | – |
| **Average** | – | 90.55 | 53.88 | 78.74 | 36.99 | 81.42 | 26.24 |

**Table 9:** Detailed Metrics of BiDexHD-IPPO for each sub-task under $\varepsilon_{\text{succ}} = \varepsilon_{\text{track}} = 0.1$.

| Action | Sub-Task Name | Train $r_1(\%)$ | Train $r_2(\%)$ | Test Comb $r_1(\%)$ | Test Comb $r_2(\%)$ | Test New $r_1(\%)$ | Test New $r_2(\%)$ |
|---|---|---|---|---|---|---|---|
| Empty | (empty, bowl, bowl) | 99.74 | 79.02 | 97.67 | 37.55 | 98.34 | 48.72 |
| | (empty, bowl, plate) | 97.90 | 75.76 | 100.00 | 62.13 | 85.00 | 9.10 |
| | (empty, cup, plate) | 99.84 | 79.47 | – | – | – | – |
| | (empty, teapot, plate) | 100.00 | 84.29 | 90.91 | 16.84 | 100.00 | 25.80 |
| | (empty, teapot, teapot) | 100.00 | 84.87 | 100.00 | 87.06 | – | – |
| Pour in some | (pour in some, cup, cup) | 100.00 | 99.70 | – | – | – | – |
| | (pour in some, cup, plate) | 89.78 | 55.95 | – | – | – | – |
| | (pour in some, cup, teapot) | 99.47 | 74.43 | – | – | – | – |
| | (pour in some, teapot, bowl) | 100.00 | 75.48 | – | – | – | – |
| | (pour in some, teapot, cup) | 95.76 | 57.23 | – | – | – | – |
| Dust | (dust, brush, bowl) | 100.00 | 91.24 | – | – | 84.34 | 32.10 |
| | (dust, brush, pan) | 100.00 | 86.08 | 100.00 | 58.09 | – | – |
| Put out | (put out, bowl, bowl) | 100.00 | 75.09 | 100.00 | 72.92 | 81.25 | 17.87 |
| | (put out, bowl, plate) | 96.98 | 77.97 | 100.00 | 84.97 | 29.41 | 7.03 |
| Skim off | (skim off, bowl, plate) | 100.00 | 73.11 | – | – | 50.00 | 8.74 |
| Smear | (smear, glue gun, plate) | 99.88 | 81.24 | – | – | – | – |
| **Average** | – | 98.71 | 78.18 | 98.37 | 59.94 | 75.48 | 21.34 |

**Table 10:** Detailed Metrics of BiDexHD-IPPO(w.o. stage-1) for each sub-task under $\varepsilon_{\text{succ}} = \varepsilon_{\text{track}} = 0.1$.

| Action | Sub-Task Name | Train $r_1(\%)$ | Train $r_2(\%)$ | Test Comb $r_1(\%)$ | Test Comb $r_2(\%)$ | Test New $r_1(\%)$ | Test New $r_2(\%)$ |
|---|---|---|---|---|---|---|---|
| Empty | (empty, bowl, bowl) | 100.00 | 73.94 | 64.63 | 49.10 | 58.08 | 45.91 |
| | (empty, bowl, plate) | 0.00 | 0.00 | 0.00 | 0.00 | 0.00 | 0.00 |
| | (empty, cup, plate) | 0.00 | 0.00 | – | – | – | – |
| | (empty, teapot, plate) | 0.00 | 0.00 | 0.00 | 0.00 | 0.00 | 0.00 |
| | (empty, teapot, teapot) | 100.00 | 83.77 | 61.32 | 45.78 | – | – |
| Pour in some | (pour in some, cup, cup) | 0.00 | 0.00 | – | – | – | – |
| | (pour in some, cup, plate) | 0.00 | 0.00 | – | – | – | – |
| | (pour in some, cup, teapot) | 0.00 | 0.00 | – | – | – | – |
| | (pour in some, teapot, bowl) | 0.00 | 0.00 | – | – | – | – |
| | (pour in some, teapot, cup) | 0.00 | 0.00 | – | – | – | – |
| Dust | (dust, brush, bowl) | 0.00 | 0.00 | – | – | 0.00 | 0.00 |
| | (dust, brush, pan) | 0.00 | 0.00 | 0.00 | 0.00 | – | – |
| Put out | (put out, bowl, bowl) | 100.00 | 71.32 | 47.65 | 31.80 | 11.02 | 6.79 |
| | (put out, bowl, plate) | 0.00 | 0.00 | 0.00 | 0.00 | 0.00 | 0.00 |
| Skim off | (skim off, bowl, plate) | 100.00 | 51.37 | – | – | 69.88 | 6.86 |
| Smear | (smear, glue gun, plate) | 0.00 | 0.00 | – | – | – | – |
| **Average** | – | 25.00 | 17.52 | 24.80 | 18.10 | 19.85 | 8.51 |

**Table 11:** Detailed Metrics of BiDexHD-IPPO(w.o. gc) for each sub-task under $\varepsilon_{\text{succ}} = \varepsilon_{\text{track}} = 0.1$.

| Action | Sub-Task Name | Train $r_1(\%)$ | Train $r_2(\%)$ | Test Comb $r_1(\%)$ | Test Comb $r_2(\%)$ | Test New $r_1(\%)$ | Test New $r_2(\%)$ |
|---|---|---|---|---|---|---|---|
| Empty | (empty, bowl, bowl) | 100.00 | 75.66 | 100.00 | 49.28 | 100.00 | 50.27 |
| | (empty, bowl, plate) | 78.68 | 45.56 | 65.00 | 16.44 | 56.67 | 10.62 |
| | (empty, cup, plate) | 52.53 | 19.67 | – | – | – | – |
| | (empty, teapot, plate) | 69.73 | 47.65 | 91.23 | 14.89 | 83.00 | 35.43 |
| | (empty, teapot, teapot) | 100.00 | 83.56 | 100.00 | 86.58 | – | – |
| Pour in some | (pour in some, cup, cup) | 100.00 | 99.71 | – | – | – | – |
| | (pour in some, cup, plate) | 98.06 | 44.36 | – | – | – | – |
| | (pour in some, cup, teapot) | 99.90 | 91.03 | – | – | – | – |
| | (pour in some, teapot, bowl) | 100.00 | 74.71 | – | – | – | – |
| | (pour in some, teapot, cup) | 94.41 | 56.88 | – | – | – | – |
| Dust | (dust, brush, bowl) | 99.78 | 88.39 | – | – | 80.15 | 27.45 |
| | (dust, brush, pan) | 94.03 | 74.31 | 84.03 | 55.75 | – | – |
| Put out | (put out, bowl, bowl) | 99.95 | 62.07 | 100.00 | 61.41 | 84.51 | 11.88 |
| | (put out, bowl, plate) | 84.15 | 66.27 | 100.00 | 80.39 | 79.59 | 18.73 |
| Skim off | (skim off, bowl, plate) | 91.40 | 70.22 | – | – | 55.26 | 4.06 |
| Smear | (smear, glue gun, plate) | 85.93 | 62.13 | – | – | – | – |
| **Average** | – | 90.53 | 66.39 | 91.47 | 52.11 | 77.03 | 22.63 |

**Table 12:** Detailed Metrics of BiDexHD-IPPO(w.o. bonus) for each sub-task under $\varepsilon_{\text{succ}} = \varepsilon_{\text{track}} = 0.1$.

| Action | Sub-Task Name | Train $r_1(\%)$ | Train $r_2(\%)$ | Test Comb $r_1(\%)$ | Test Comb $r_2(\%)$ | Test New $r_1(\%)$ | Test New $r_2(\%)$ |
|---|---|---|---|---|---|---|---|
| Empty | (empty, bowl, bowl) | 100.00 | 52.77 | 98.91 | 44.44 | 96.49 | 24.22 |
| | (empty, bowl, plate) | 98.95 | 66.06 | 99.51 | 63.93 | 84.87 | 8.59 |
| | (empty, cup, plate) | 99.77 | 43.87 | – | – | – | – |
| | (empty, teapot, plate) | 100.00 | 83.39 | 92.66 | 24.63 | 86.70 | 26.44 |
| | (empty, teapot, teapot) | 100.00 | 84.03 | 99.61 | 85.76 | – | – |
| Pour in some | (pour in some, cup, cup) | 100.00 | 98.99 | – | – | – | – |
| | (pour in some, cup, plate) | 89.91 | 31.76 | – | – | – | – |
| | (pour in some, cup, teapot) | 100.00 | 86.55 | – | – | – | – |
| | (pour in some, teapot, bowl) | 76.43 | 11.27 | – | – | – | – |
| | (pour in some, teapot, cup) | 100.00 | 67.22 | – | – | – | – |
| Dust | (dust, brush, bowl) | 99.77 | 86.93 | – | – | 88.34 | 35.45 |
| | (dust, brush, pan) | 99.44 | 72.85 | 95.41 | 46.95 | – | – |
| Put out | (put out, bowl, bowl) | 100.00 | 62.52 | 100.00 | 68.21 | 95.45 | 16.13 |
| | (put out, bowl, plate) | 98.80 | 75.67 | 100.00 | 84.37 | 48.82 | 5.16 |
| Skim off | (skim off, bowl, plate) | 100.00 | 58.47 | – | – | 45.03 | 6.66 |
| Smear | (smear, glue gun, plate) | 99.70 | 84.10 | – | – | – | – |
| **Average** | – | 97.67 | 66.65 | 98.01 | 59.76 | 77.96 | 17.52 |

## C.2 RESULTS OF STUDENT LEARNING

Tables below record the detailed evaluation results for each task category. The last row of each table shows the average results over all sub-tasks in all task categories. '–' in the table represents the absence of testing tasks.

**Table 13:** Detailed Metrics of BiDexHD-PPO+DAgger for each task category under $\varepsilon_{\text{succ}} = \varepsilon_{\text{track}} = 0.1$.

| Action | Train $r_1(\%)$ | Train $r_2(\%)$ | Test Comb $r_1(\%)$ | Test Comb $r_2(\%)$ | Test New $r_1(\%)$ | Test New $r_2(\%)$ |
|---|---|---|---|---|---|---|
| Dust (2) | 95.42 | 72.49 | 72.41 | 19.13 | 94.92 | 40.79 |
| Empty (5) | 96.61 | 56.95 | 70.18 | 24.58 | 90.91 | 27.63 |
| Put out (2) | 98.04 | 52.33 | 97.52 | 56.31 | 88.41 | 29.75 |
| Pour in some (5) | 91.73 | 42.96 | – | – | – | – |
| Skim off (1) | 97.60 | 71.54 | – | – | 42.19 | 20.79 |
| Smear (1) | 99.40 | 72.46 | – | – | – | – |
| **Average** | 95.35 | 55.82 | 76.75 | 30.42 | 86.34 | 30.00 |

**Table 14:** Detailed Metrics of BiDexHD-IPPO+DAgger(K=5) for task category under $\varepsilon_{\text{succ}} = \varepsilon_{\text{track}} = 0.1$.

| Action | Train $r_1(\%)$ | Train $r_2(\%)$ | Test Comb $r_1(\%)$ | Test Comb $r_2(\%)$ | Test New $r_1(\%)$ | Test New $r_2(\%)$ |
|---|---|---|---|---|---|---|
| Dust (2) | 100.00 | 86.91 | 100.00 | 49.94 | 100.00 | 48.37 |
| Empty (5) | 99.04 | 73.20 | 87.13 | 36.62 | 96.61 | 61.96 |
| Put out (2) | 98.22 | 71.89 | 100.00 | 76.43 | 93.63 | 42.40 |
| Pour in some (5) | 100.00 | 72.31 | – | – | – | – |
| Skim off (1) | 98.78 | 71.59 | – | – | 77.58 | 45.71 |
| Smear (1) | 99.70 | 76.69 | – | – | – | – |
| **Average** | 99.38 | 74.59 | 92.85 | 48.43 | 94.79 | 53.71 |

**Table 15:** Detailed Metrics of BiDexHD-IPPO+DAgger(K=5) for each task category under $\varepsilon_{\text{succ}} = \varepsilon_{\text{track}} = 0.075$.

| Action | Train $r_1(\%)$ | Train $r_2(\%)$ | Test Comb $r_1(\%)$ | Test Comb $r_2(\%)$ | Test New $r_1(\%)$ | Test New $r_2(\%)$ |
|---|---|---|---|---|---|---|
| Dust (2) | 99.64 | 75.79 | 97.06 | 22.34 | 96.97 | 28.81 |
| Empty (5) | 97.26 | 63.39 | 61.11 | 10.62 | 96.59 | 48.16 |
| Put out (2) | 94.44 | 60.38 | 100.00 | 61.64 | 62.50 | 20.29 |
| Pour in some (5) | 100.00 | 68.38 | – | – | – | – |
| Skim off (1) | 98.53 | 60.74 | – | – | 79.17 | 37.23 |
| Smear (1) | 99.40 | 67.15 | – | – | – | – |
| **Average** | 98.27 | 66.19 | 77.74 | 24.56 | 88.11 | 37.62 |

**Table 16:** Detailed Metrics of BiDexHD-IPPO+DAgger(K=5) for each task category under $\varepsilon_{\text{succ}} = \varepsilon_{\text{track}} = 0.05$.

| Action | Train $r_1(\%)$ | Train $r_2(\%)$ | Test Comb $r_1(\%)$ | Test Comb $r_2(\%)$ | Test New $r_1(\%)$ | Test New $r_2(\%)$ |
|---|---|---|---|---|---|---|
| Dust (2) | 97.51 | 58.09 | 52.38 | 5.22 | 84.62 | 14.20 |
| Empty (5) | 94.52 | 50.47 | 27.78 | 6.65 | 88.89 | 21.71 |
| Put out (2) | 93.05 | 41.78 | 100.00 | 36.75 | 56.25 | 10.49 |
| Pour in some (5) | 100.00 | 59.37 | – | – | – | – |
| Skim off (1) | 98.75 | 42.11 | – | – | 69.39 | 13.94 |
| Smear (1) | 97.44 | 50.18 | – | – | – | – |
| **Average** | 96.87 | 52.58 | 49.30 | 13.02 | 79.56 | 17.19 |

**Table 17:** Detailed Metrics of BiDexHD-IPPO+DAgger(K=2) for each task category under $\varepsilon_{\text{succ}} = \varepsilon_{\text{track}} = 0.1$.

| Action | Train $r_1(\%)$ | Train $r_2(\%)$ | Test Comb $r_1(\%)$ | Test Comb $r_2(\%)$ | Test New $r_1(\%)$ | Test New $r_2(\%)$ |
|---|---|---|---|---|---|---|
| Dust (2) | 99.81 | 86.47 | 98.33 | 48.19 | 99.79 | 52.57 |
| Empty (5) | 98.42 | 73.13 | 88.54 | 36.51 | 97.30 | 55.76 |
| Put out (2) | 98.37 | 70.56 | 100.00 | 79.25 | 91.41 | 39.62 |
| Pour in some (5) | 98.47 | 74.59 | – | – | – | – |
| Skim off (1) | 98.55 | 70.32 | – | – | 74.87 | 40.77 |
| Smear (1) | 100.00 | 77.14 | – | – | – | – |
| **Average** | 98.71 | 75.01 | 93.26 | 48.60 | 94.38 | 50.39 |

**Table 18:** Detailed Metrics of BiDexHD-IPPO+DAgger(K=1) for each task category under $\varepsilon_{\text{succ}} = \varepsilon_{\text{track}} = 0.1$.

| Action | Train $r_1(\%)$ | Train $r_2(\%)$ | Test Comb $r_1(\%)$ | Test Comb $r_2(\%)$ | Test New $r_1(\%)$ | Test New $r_2(\%)$ |
|---|---|---|---|---|---|---|
| Dust (2) | 99.89 | 85.90 | 99.19 | 45.88 | 100.00 | 47.51 |
| Empty (5) | 98.55 | 73.04 | 86.13 | 39.56 | 98.44 | 59.65 |
| Put out (2) | 97.90 | 72.42 | 100.00 | 75.80 | 90.91 | 38.03 |
| Pour in some (5) | 98.78 | 74.90 | – | – | – | – |
| Skim off (1) | 98.94 | 71.45 | – | – | 72.65 | 40.66 |
| Smear (1) | 99.87 | 78.54 | – | – | – | – |
| **Average** | 98.81 | 75.40 | 92.11 | 49.02 | 94.67 | 51.00 |

**Table 19:** Detailed Metrics of BiDexHD-IPPO+DAgger(K=0) for each task category under $\varepsilon_{\text{succ}} = \varepsilon_{\text{track}} = 0.1$.

| Action | Train $r_1(\%)$ | Train $r_2(\%)$ | Test Comb $r_1(\%)$ | Test Comb $r_2(\%)$ | Test New $r_1(\%)$ | Test New $r_2(\%)$ |
|---|---|---|---|---|---|---|
| Dust (2) | 100.00 | 86.27 | 98.21 | 44.53 | 98.72 | 42.01 |
| Empty (5) | 95.77 | 64.37 | 90.57 | 35.44 | 96.20 | 59.15 |
| Put out (2) | 98.08 | 72.92 | 100.00 | 76.74 | 90.84 | 39.50 |
| Pour in some (5) | 99.01 | 73.35 | – | – | – | – |
| Skim off (1) | 98.39 | 69.76 | – | – | 79.44 | 33.94 |
| Smear (1) | 99.70 | 76.63 | – | – | – | – |
| **Average** | 98.01 | 72.09 | 94.36 | 46.64 | 93.96 | 49.27 |

# D ADDITIONAL VISUALIZATIONS

Figures below visualize samples of bimanual human demonstrations and policy deployment of constructed bimanual dexterous manipulation tasks.

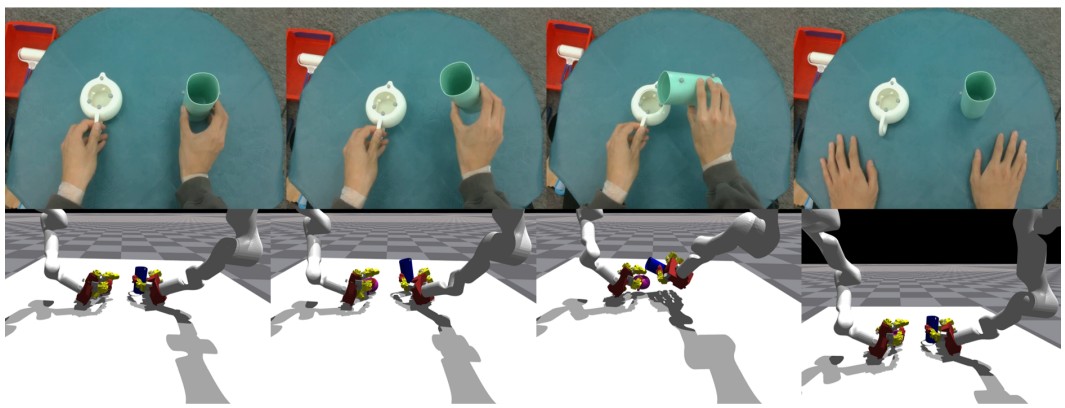

**Figure 5:** Task visualization of (pour in some, cup, teapot).

**Figure 6:** Task visualization of (empty, bowl, bowl).

# E    COMPARISONS WITH PREVIOUS WORK

**PGDM** Dasari et al. (2023) plans with given pre-grasp pose and trains policies to track the human trajectories from object trajectories via reinforcement learning. Although the methodology shows some similarity to BiDexHD and diverse single-hand manipulation tasks are also addressed, there are significant differences between PGDM and BiDexHD:

- Stage Division. PGDM divides the task into distinct stages: planning a pre-grasp pose (reaching stage) and learning to grasp and move through reinforcement learning (grasping and moving stages). Their planning-based reaching stage is limited to performing hand-reaching behaviors, while our BiDexHD can perform general, contact-rich behaviors in the RL-based alignment stage. BiDexHD employs unified reinforcement learning, starting with aligning both hands and objects to a ready state (alignment stage), followed by trajectory tracking (tracking stage). This design allows BiDexHD to flexibly learn diverse skills like twisting and pushing, going beyond simple reaching and grasping. Once both hands securely hold the objects, they maintain their relative states and learn to track desired poses with ease. Our design properly strikes a balance between policy quality and training difficulty.

- Data requirement. PGDM relies on human-annotated pre-grasp poses in the TCDM benchmark for planning. BiDexHD uses RL to learn to align the simulation with the datasets without extra annotations, enhancing its scalability.

- Application Scope. PGDM primarily focuses on single Adroit Hand grasping in Mujoco simulations, while BiDexHD extends to more complex bimanual arm-hand systems and diverse manipulation tasks in highly parallelized IsaacGym simulations.

**Dexcap** Wang et al. (2024a) proposes a novel motion capture and vision-based data collection system for bimanual task learning via imitation. It is worth noting that data collected by Dexcap alone is insufficient to derive feasible policies. Further human-in-the-loop finetuning is necessary to incorporate more kinematics and dynamics. In contrast, BiDexHD uses online reinforcement learning with a general reward function to learn diverse bimanual skills from object motion capture data through trial and error, without additional fine-tuning.

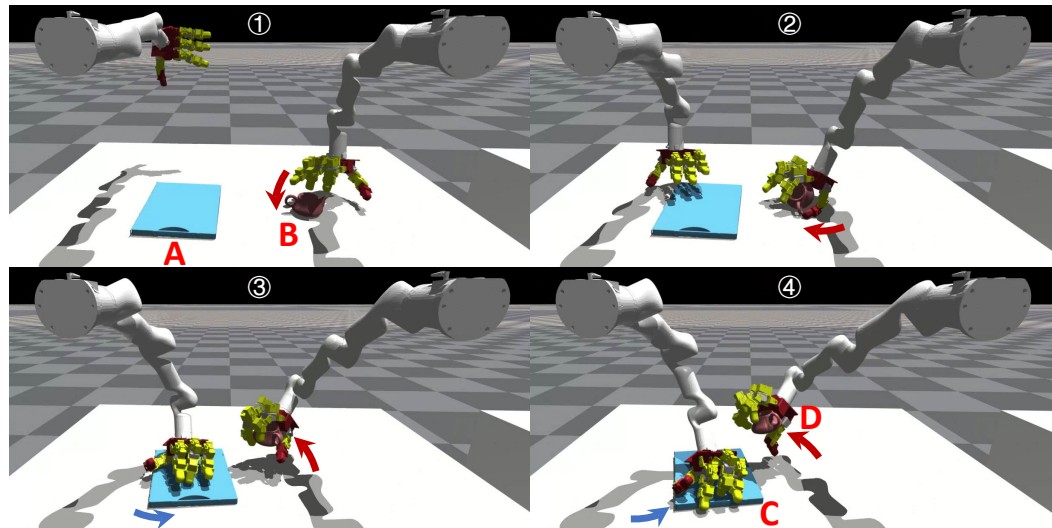

**Figure 7:** The alignment stage of (empty, teapot, plate) task. We target to align the state described in the 4th frame with the initial state of the demonstrated trajectory. During the alignment stage in BiDexHD, the right hand is supposed to learn to approach, grasp, re-orient, and lift the teapot and the left hand needs to learn to approach and push the plate via reinforcement learning. Object trajectory tracking starts only after the simulation-dataset alignment has been successfully completed, and no additional trajectory information is provided before this in the dataset. Therefore, it is hard to realize the intensive hand-object interaction only through planning-based methods like PGDM Dasari et al. (2023).

