# OpenReview forum: "Learning Diverse Bimanual Dexterous Manipulation Skills from Human Demonstrations"
_ICLR.cc/2025/Conference — Submitted to ICLR 2025_

### Official Review · Reviewer_CkYS · 2024-11-02

**Soundness:** 3
**Presentation:** 3
**Contribution:** 2
**Rating:** 5
**Confidence:** 4

**Summary:**

This paper introduces an approach to learn multi-task bimanual manipulation policies through teacher-student training from human demonstrations. Before training, it uses existing bimanual dataset of human videos to extract human poses and other relevant information  for task construction. During training, It first trains a RL policy to grab the tool and object by two hands separately with multi-stage reward. Then it uses reward of tracking object poses to train a state-based RL teacher policy. After getting two-stage policies, it distills the policy to a point-cloud based policy with DAgger.

**Strengths:**

1. This paper proposes very detailed description of the pipeline and outlines each component and different stages
2. The learned multi-task policy can work on six categories of different bimanual tasks and outperforms the baselines.
3. The paper also ablates the different components in the whole pipeline to highlight the importance of each component.

**Weaknesses:**

1. The paper uses the metrics of difference between object pose  and the reference object pose along the trajectory as the success criteria.  The proposed method has better performance in this metrics than the baselines. However, from the video shown on the website. The policy looks very jerky and sometimes the hand is in touch of the table. Therefore, this reduces the plausibility of this learned policy to transfer to the real robot.  It might be helpful to use other metrics to evaluate the policy such as number of collisions, the smoothness of the policy, etc. to have a comprehensive review of the paper.
2. The paper uses the teacher-student policy distillation to learn the bimanual policy which is commonly used for single dexterous hand manipulation. Extracting human poses from human videos are also common in prior work. Therefore, it is unclear to me what is the key innovation and contribution of this paper.
3. The paper introduces the behavior cloning (BC) baseline but doesn’t mention which algorithm they are using. Currently there are several popular BC methods including Diffusion Policy, BC Transformer, ACT, etc. The paper should include more details of the BC policy including the algorithm, the hyperparameters and the demonstration dataset size.

**Questions:**

1. There is one recent bimanual manipulation policies from point-cloud, DexCap[1] with imitation learning. That shows good results on learning tool use with bimanual manipulation. It would be interesting to see how their point-cloud based policy works in these tasks.
2. It is a little surprising to see the IPPO has better results than PPO because some of bimanual manipulation tasks may need coordination. Could the authors provide more explanations about why IPPO behaves better than PPO.
3. The paper only compares PPO, IPPO and some ablation of different components of the IPPO. However, in the previous bimanual dexterous manipulation paper[2], they compare a variety of RL, MARL methods. It would make the paper stronger if the proposed method is also compared against MARL methods.
4. What are the common failure modes of the baselines? It would be interesting to see the comparison between the videos of the  proposed methods and the baselines. Currently, the website only has videos of the proposed methods.
5. The paper already extracts the human poses from the bimanual demonstration dataset and can retarget from human poses to robot hand joint. Why not also use this data to add tracking reward not only to object pose but also to robot hand joint?
6. Adding the future object positions into the observations seems not very practical in real world. I am wondering if the authors have any insight about how to get future object positions to feed into the policy when deployed in the real world.

[1]Wang, Chen, et al. "Dexcap: Scalable and portable mocap data collection system for dexterous manipulation." In proceedings of Robotics: Science and System, 2024.

[2]Y. Chen et al., "Bi-DexHands: Towards Human-Level Bimanual Dexterous Manipulation," in IEEE Transactions on Pattern Analysis and Machine Intelligence, vol. 46, no. 5, pp. 2804-2818, May 2024,

---

> ### Author Response · Authors · 2024-11-22
> **Reply for Reviewer CkYS**
>
> Thanks for the careful review and constructive suggestions. We want to address the questions and concerns below.
>
> `Q1`: "Use of other metrics to evaluate the quality of policy"
>
> `A1`: In `Author Rebuttal 4` we explain the reasons for jerky motion and emphasize that considering BiDexHD is the first preliminary attempt towards scalable bimanual skill learning from diverse constructed tasks we **prioritize achieving high task completion rates** for challenging bimanual dexterous tasks. In other words, other properties are not the central goals. Of course, for additional safety concerns, we would like to add regularization terms to penalize joint angles, velocities, accelerations, and jerks for sim-to-real deployment. Please refer to `Author Rebuttal 4` for more details.
>
> `Q2`: "Key innovation and contribution"
>
> `A2`: We would like to emphasize that **BiDexHD is the first framework to (1) automatically construct diverse bimanual tasks from human demonstrations without task-specific design, and (2) solve them using a general reward function in a unified manner.** Please refer to `Author Rebuttal 2` for more explanations and comparisons.
>
> `Q3`: "BC details"
>
> `A3`: Please refer to `Author Rebuttal 3` and `Appendix B.5` for a detailed explanation. We analyze that **data quality and quantity** account for the bad performance of BC, regardless of Diffusion or ACT training methods.
>
> `Q4`: "Recent work DexCap"
>
> `A4`: Actually, In BiDexHd we distill multiple state-based expert policies into single point-cloud-based policies to solve tasks with similar behavior. We conduct more comparisons with Dexcap [1] and other baselines in `Author Rebuttal 2`.
>
> `Q5`: "Performance comparison between IPPO and PPO"
>
> `A5`: We provide more explanations in `Author Rebuttal 6`.
>
> `Q6`: "Whether compare against MARL methods like [2]"
>
> `A6`: Bi-dexhands [2] is an RL benchmark for bimanual dexterous manipulation. To verify wide task diversity and different levels of task difficulty, it benchmarks many kinds of RL / MARL / Offline RL / Multi-task RL / Meta RL methods. **BiDexHD is the first preliminary attempt featuring scalable bimanual skill learning from diverse automatically constructed tasks**. No matter what variants are,  "BiDexHD-X" addresses two core points mentioned in `A2`. We thank the reviewer for the kind reminder and would like to incorporate multi-agent RL algorithms in future work.
>
> `Q7`: "Failure modes of the baselines"
>
> `A7`: We demonstrate some videos of the BC baseline on our project page [BiDexHD](https://sites.google.com/view/bidexhd) (in the second to last section).
>
> `Q8`: "Why not use the retargeted data to add tracking reward"
>
> `A8`: As explained in the second failure reason of BC in `Author Rebuttal 3`, the retargeted robot joint angles are **non-expert** and tend to **exhibit inconsistent kinematics and unreal dynamics**. Therefore, it may not sound like a good idea to use these inaccurate reward signals. Besides, as we mainly concentrate on object-centric trajectories, different behaviors are welcome and acceptable. In other words, as one task comes from one trajectory, it is not necessary for the bimanual system to only imitate a single behavior pattern. By only focusing on the object pose transformation, it is possible to learn a better policy that shows some difference to the human demonstration but is more suitable for robotic dexterous hands to solve the task. Thus, in BiDexHd we only use retargeted results for visualization and invalid task identification.
>
> `Q9`: "How to get future object positions in the real world"
>
> `A9`: As explained in `Author Rebuttal 5`, we have come up with two solutions to estimate future poses of real objects:
>
>   - Use **large multimodal models** to generate valid future trajectories according to historical observations and trajectories.
>   - Train an **object motion prediction model** from various object manipulation datasets.
>
> Thanks again for the review! We will implement the feedback in the final version of this paper. Further comments are welcome!
>
> **Reference**
>
> [1] Wang, Chen, et al. "Dexcap: Scalable and portable mocap data collection system for dexterous manipulation." *ArXiv 2024*.
>
> [2]Chen, Yuanpei, et al. "Bi-dexhands: Towards human-level bimanual dexterous manipulation." *TPAMI 2023.*

---

> ### Author Response · Authors · 2024-11-25
> **For Reviewer CkYS**
>
> Thanks again for your careful review! Here, we respond to your comments and address the issues. We hope to hear back from you! If you have further questions, feel free to let us know, and ***\*we are more than happy to answer additional questions\****. If you feel that our rebuttal has addressed your concerns, we would be grateful if you would consider ***\*revising your score in response\****.

---

> > ### Comment · Reviewer_CkYS · 2024-11-27
> > **Response to the authors**
> >
> > I would like to thank the authors for their added explanations of the failure modes from the BC policies and updated videos on the website. This helps readers understand more about the strength of the proposed methods over BC policies. I also appreciate the authors' efforts for summarizing and categorizing prior work in bimanual dexterous manipulation in the paper.  Therefore, I would raise the presentation score of this paper.
> >
> > My major concern is still the sim2real challenge as shared by other reviewers too. Although the authors proposed some solutions for reducing the jerkiness of motions by adding the reward to encourage the smoothness of the motions, it is still not clear to me if adding reward can fundamentally eliminate this issue and provide smooth transition to the real bimanual hands. Second, I am still concerning about metrics, the author argues that they are prioritizing high task completion rate and other goals can be left for future work. However, I am not sure if having jerky and highly unstable motions but only satisfying some pre-defined success criterion is an actual task completion. Lastly, regarding the estimating future object trajectories, the author mentions two related work. [5] is used to estimate human motion data. Although these are possible potential approaches to have planned object trajectories, it is non-trivial to get this work reliably to have relatively accurate future object trajectories, (i.e., training large models on large object datasets, etc.) so I would say this assumption is one limitation of this work.
> >
> > Other questions have been addressed by the authors' detailed response. Because of this major concern, I would still keep my current score.

---

> > > ### Author Response · Authors · 2024-11-27
> > > **Further rebuttal for Reviewer CkYS [1/2]**
> > >
> > > Thank you for your reply. We would like to address your remaining concerns:
> > >
> > > `Q10`: Whether adding a penalty reward can reduce the jerkiness of motions.
> > >
> > > `A10`: **It is a common practice in recent reinforcement learning (RL)-based approaches [1,2,3], particularly for dexterous manipulation tasks, to incorporate a reward term that penalizes the norm of normalized robot actions or proprioceptive states**, such as joint angles, velocities, accelerations, and jerks. This approach has proven effective in stabilizing motion and is also widely adopted across different robotic embodiments, including quadrupeds [9] and humanoids [10]. So many RL-related studies provide evidence that introducing a penalty reward can significantly reduce motion instability, mitigate jerky behavior, and improve energy efficiency. For real-world deployment, where safety and smoothness are critical, we will balance the weight of the penalty term alongside other rewards to ensure a smooth and controlled performance.
> > >
> > > `Q11`: About metrics.
> > >
> > > `A11`: **Many recent studies [4,5,6] focusing on challenging bimanual dexterous manipulation tasks use the success rate as a primary metric to evaluate the effectiveness of bimanual policies**. This metric is crucial because it directly reflects whether objects are successfully moved to the target location in object-centric tasks. In BiDexHD, we follow previous work in using a similar task completion rate metric, specifically for phase one ($r_1$) and phase two ($r_2$), which are correlated to the given object trajectories.
> > >
> > > `Q12`: About future object trajectories.
> > >
> > > `A12`:
> > >
> > > 1. In **Section 5.4**, we demonstrate that $r_2$ experiences only slight declines (2.5%) on trained tasks and an average of 3.1% on all unseen tasks, even when future conditioned steps are masked. This suggests that **pure imitation from state-based policies, without relying on future-conditioned steps, is sufficient for a vision-based policy to achieve acceptable performance**. For real-world deployment, a policy conditioned only on robot proprioception and point clouds can just achieve competitive performance with appropriate hyperparameter tuning.
> > > 2. Furthermore, BiDexHD is well-positioned as an effective approach for low-level dexterous skill learning.
> > >    - Similar work such as **Omnigrasp** [7], which also focuses on low-level control, **incorporates future object trajectories into the policy**. We adopt a similar approach in BiDexHD, and our empirical results indicate that future steps provide valuable, fine-grained information such as motion and intention, which aids in more precise tracking.
> > >    - **Predicting future object trajectories falls under the domain of high-level planning, which is beyond the scope of our current study on low-level control**. Object trajectories are more closely tied to the scene and task properties, rather than the dexterous actions of the hands. Therefore, we can easily integrate existing generalizable object motion prediction models (such as [8]) which are trained on large datasets of object interactions, to produce future trajectories for real-world policy deployment. We do not need to train such a prediction model on our limited object data, and this should be the research focus of high-level foundation models.
> > >
> > > We believe in this paper we propose a unified and scalable framework BiDexHD towards the underexplored problem of generally  learning diverse bimanual dexterous manipulation skills from single human demonstration. Other reviewers also express an overall positive attitude towards the overall contributions of BiDexHD especially for **"automatic task construction" and "general reward function"**. Regarding concerns about sim-to-real transfer, the challenges of bimanual sim-to-real, including simulation gaps, control gaps, safety concerns, and smoothness, are widely acknowledged by the community. Much work remains to be done to fully address these issues. We hope that you will consider the promising results from our simulations and the potential for scaling up this work in the future. We kindly hope that you reconsider the score, and we would be happy to address any further concerns.

---

> > > > ### Author Response · Authors · 2024-11-27
> > > > **Further rebuttal for Reviewer CkYS [2/2]**
> > > >
> > > > **Reference**
> > > >
> > > > [1] Wu, Tianhao, et al. "Unidexfpm: Universal dexterous functional pre-grasp manipulation via diffusion policy." *Arxiv 2024*.
> > > >
> > > > [2] Wan, Weikang, et al. "Unidexgrasp++: Improving dexterous grasping policy learning via geometry-aware curriculum and iterative generalist-specialist learning." *ICCV 2023*.
> > > >
> > > > [3] Wu, Tianhao, et al. "Learning score-based grasping primitive for human-assisting dexterous grasping." *NeurIPS 2024*.
> > > >
> > > > [4] Wang, Chen, et al. "Dexcap: Scalable and portable mocap data collection system for dexterous manipulation." *ArXiv 2024*.
> > > >
> > > > [5] Wang, Shiyao, et al. "Physics-aware iterative learning and prediction of saliency map for bimanual grasp planning."  *CAGD 2024*.
> > > >
> > > > [6] Zhang, Hui, et al. "ArtiGrasp: Physically plausible synthesis of bi-manual dexterous grasping and articulation." *3DV 2024*.
> > > >
> > > > [7] Luo, Zhengyi, et al. "Grasping diverse objects with simulated humanoids." *ArXiv 2024*.
> > > >
> > > > [8] Xiao, Changcheng, et al. "Motiontrack: Learning motion predictor for multiple object tracking." *Neural Networks 2024.*
> > > >
> > > > [9] Shafiee, Milad, Guillaume Bellegarda, and Auke Ijspeert. "Manyquadrupeds: Learning a single locomotion policy for diverse quadruped robots." *ICRA 2024*.
> > > >
> > > > [10] Dao, Jeremy, Helei Duan, and Alan Fern. "Sim-to-real learning for humanoid box loco-manipulation." *ICRA 2024*.

---

> > > > > ### Author Response · Authors · 2024-11-28
> > > > > **For Reviewer CkYS**
> > > > >
> > > > > Here, we respond to your comments and address the issues. If you have further questions, feel free to let us know, and *we are more than happy to answer additional questions*. If you feel that our rebuttal has addressed your concerns, we would be grateful if you would consider *revising your score in response*. We hope to hear back from you!

---

### Official Review · Reviewer_68Pu · 2024-11-03

**Soundness:** 3
**Presentation:** 4
**Contribution:** 2
**Rating:** 5
**Confidence:** 4

**Summary:**

This paper proposes an approach that learns bimanual dexterous manipulation from human demonstration dataset. The approach constructs corresponding tasks from existing bimanual dataset and applies teacher-student learning on the constructed tasks. The task construction part includes data preprocessing which converts human hand poses to LEAP hand pose and simulation initialization which initializes corresponding objects in the simulation environment according to the demonstration. Their approach also decomposes the teacher policy learning into two stages: 1) training the policy to matching the objects and hand poses with the first time step of the demonstration trajectory from initial pose and 2) tracking the reference object trajectory. The authors compare performance of different RL algorithms in teacher policy learning and different IL algorithms in vision-based policy distillation. They also ablate several design choices in teacher-student policy learning and demonstrate improved performance over other baselines.

**Strengths:**

By utilizing human demonstration dataset, BiDexHD learns bimanual dexterous policy in a scalable way. Unlike some prior works that is only limited to learning a specific task, BiDexHD is capable of learning many bimanual dexterous skills, meanwhile avoiding the effort of complex reward shaping for individual tasks. The author compares BiDexHD against ablated baselines and demonstrates its high performance over the baselines and competitive generalization capabilities.

**Weaknesses:**

1). Although BiDexHD is able to learn many bimanual tasks, it is tailored to learning one category of tasks: one hand holding the tool and another hand holding the object. It seems the framework is not able to deal with tasks that require in hand manipulation of either hand, or bimanual manipulation of a single object. For example, hand over and open bottle cap.

2). Description of Fig. 2 mentioned the tool and object are initialized at a fixed pose, but in real world application, the manipulated objects are seldom initialized at a fixed pose. The policy are not trained on a randomized initial pose.

3). Learning bimanual dexterous skills is extremely challenging, hence it is still unclear if the learned policies are able to be deployed on real-world hardwares.

4). In Sec. 5.2, the metric $r_2$ might not be a complete metric for task completion, because the task might still be completed while the hands fail to track the demonstration. For instance, when the tool and object both move up the same distance, while their relative pose keeps constant, the task is still completed.

5). The use of object trajectory tracking in reinforcement learning for dexterous manipulation is not particularly novel, as evidenced by prior works such as [1][2][3][4]. It would add depth to the discussion if the authors could address why existing trajectory-tracking methods cannot be directly adapted for bimanual dexterous tasks. While I understand that bimanual tasks indeed expand the observation and action spaces significantly, it would be helpful to know if there are additional, perhaps more nuanced, challenges that prevent a straightforward adaptation of these methods.

[1] Han, Yunhai, et al. "Learning Prehensile Dexterity by Imitating and Emulating State-Only Observations." IEEE Robotics and Automation Letters (2024).
[2] Dasari, Sudeep, Abhinav Gupta, and Vikash Kumar. "Learning dexterous manipulation from exemplar object trajectories and pre-grasps." 2023 IEEE International Conference on Robotics and Automation (ICRA). IEEE, 2023.
[3] Guzey, Irmak, et al. "Bridging the Human to Robot Dexterity Gap through Object-Oriented Rewards." arXiv preprint arXiv:2410.23289 (2024).
[4] Chen, Yuanpei, et al. "Object-Centric Dexterous Manipulation from Human Motion Data." 8th Annual Conference on Robot Learning.

**Questions:**

1). The learned policy in the video includes a lot of jerking motion, is it possible to add some action penalty term in RL to smooth the motion?

2). Sec 4.2 mentioned identifying and removing invalid tasks to build up a complete task set. I am wondering how to identify invalid tasks after initialization.

3). Could you provide analysis on why BiDexHD-PPO outperforms BiDexHD-IPPO on teacher learning?

4). There has also been some work that learns bimanual dexterous policy from demonstration data. What is the difference between BiDexHD and other bimanual dexterous manipulation papers, for instance, DexCap?

5). Some discussion if the authors could address why existing trajectory-tracking methods ([1-4]) cannot be directly adapted for bimanual dexterous tasks.

---

> ### Author Response · Authors · 2024-11-22
> **Reply for Reviewer 68Pu**
>
> Thanks for the careful review and valuable feedback! We are encouraged that two main points "diverse dexterous skills learning from demonstrations" and "avoiding complex reward shaping for individual tasks" are delivered. We want to address the questions and concerns below.
>
> `Q1`: "Tasks about in-hand manipulation of either hand or bimanual manipulation of a single object"
>
> `A1`: In BiDexHD, we primarily focus on bimanual rigid-body-centric manipulation tasks, i.e. aligning the sequential pose transformations of objects in the simulation to those in the datasets. Therefore, BiDexHD does not excel at dealing with in-hand manipulation tasks. For bimanual manipulation of a single object, we have supplemented experiments in `Author Rebuttal 1`.
>
> `Q2`: "Fixed pose initialization"
>
> `A2`: We feel sorry that we did not well express the initial settings in the figure caption. "Fix poses" should be substituted for "poses sampled from a fixed Gaussian distribution centered at a fixed value with added small noise". We have updated the caption of Fig. 2 in the paper.
>
> `Q3`: "Deployment on real-world hardware"
>
> `A3`: We consider several major challenges and feasible solutions for the real-world deployment of BiDexHD in `Author Rebuttal 5`.
>
> `Q4`: "The metric of $r_2$"
>
> `A4`: We would claim that $r_2$ measures how many steps both the object and tool match the given trajectory. It is not designed to encourage keeping their relative positions unchanged. At each timestep $t$ during the tracking stage, we encourage the pose of the object and the pose of the tool to both get close to their desired poses.
>
> `Q5`: "Whether trajectory-tracking methods can be adapted for bimanual dexterous tasks"
>
> `A5`: We would like to emphasize that BiDexHD indeed modifies from trajectory-tracking methods. As is explained in `Author Rebuttal 2`, to flexibly learn diverse contact-rich skills, we specially design different alignment stages and tracking stages for bimanual manipulation tasks. Please refer to `Author Rebuttal 2` for a detailed comparison with more methods. We want to address **the core contributions of BiDexHD mainly lie in (1) automatically constructing diverse bimanual tasks from human demonstrations without task-specific design, and (2) solving them using a general reward function in a unified manner**. Based on the above insights, BiDexHD is the first one to be capable of scaling diverse bimanual object-centric trajectory-tracking tasks, even if the high-dimensional action space is challenging.
>
> `Q6`: "Jerking motion"
>
> `A6`: We explain the possible reasons for jerky motions in `Author Rebuttal 4` and mention adding regularization terms to penalize joint angles, velocities, accelerations, and jerk to the ultimate objective is hopefully better for training a smoother policy.
>
> `Q7`: "Identify invalid tasks after initialization."
>
> `A7`: we filter out invalid tasks mainly by checking if a task can be completed by robotic dexterous hands. Since each task is built from a human trajectory, after retargeting, we would check (1) whether the retargeted motion is continuous and valid and (2) whether an object can reach its desired pose without collision or other physics problems.
>
> `Q8`: "Performance comparison between BiDexHD-IPPO and BiDexHD-PPO"
>
> `A8`: Generally speaking, BiDexHD-IPPO outperforms BiDexHD-PPO in most cases. We explain the difference in detail in `Author Rebuttal 6`.
>
> `Q9`: "Difference between BiDexHD and other bimanual dexterous manipulation work"
>
> `A9`: We explain the difference in detail in `Author Rebuttal 2`.
>
> `Q10`: "Whether existing trajectory-tracking methods can be adapted for bimanual dexterous tasks."
>
> `A10`: Please refer to `A5`.
>
> Thanks again for the review! We will implement the feedback in the final version of this paper. Further comments are welcome!

---

> ### Author Response · Authors · 2024-11-25
> **For Reviewer 68Pu**
>
> Thanks again for your careful review! Here, we respond to your comments and address the issues. We hope to hear back from you! If you have further questions, feel free to let us know, and ***\*we are more than happy to answer additional questions\****. If you feel that our rebuttal has addressed your concerns, we would be grateful if you would consider ***\*revising your score in response\****.

---

> ### Author Response · Authors · 2024-12-02
> **For Reviewer 68Pu**
>
> Dear Reviewer 68Pu,
>
> Considering the discussion ends soon, we were wondering whether our responses address your concerns. If there are remaining concerns, **we would be delighted to have further discussion**. If our responses have addressed your concerns, we hope **the reviewer will be willing to raise the score**. Thanks again for your time and efforts in reviewing and improving our work.
>
> Sincerely,
>
> Submission1272 Authors

---

### Official Review · Reviewer_MLbc · 2024-11-03

**Soundness:** 3
**Presentation:** 3
**Contribution:** 2
**Rating:** 5
**Confidence:** 4

**Summary:**

The authors propose BiDexHD for learning bimanual manipulation policies starting from human demonstrations. From human demonstrations, BiDexHD extracts human and object poses and defines the task based on them. During the policy training phase, BiDexHD first learns state-based policies that first aligns the hands and objects to the desired poses from human demos using RL and carefully designed reward functions, and then distills them into vision-based policies using DAgger. Experiments are based on the TACO dataset including six task categories. Ablation studies show the importance of aligning phase, and the main results show improvement over BC-only baseline.

**Strengths:**

I appreciate the authors carefully detailing the approach including the different tricks applied for setting up simulation environments based on human demos, and also the carefully designed reward functions. I don’t find major issues with the notations.

The overall approach is intuitive. It is clever to use human demos as an implicit representation of the task (despite not novel, see below), and then design RL training around it.

The experiment results are generally positive. The authors show the benefits of using IPPO instead of PPO in such a decentralized setup. The improvement over BC is also clear (despite the concern about the experiment setup).

**Weaknesses:**

My main concern of the paper is the lack of technical contribution over existing work. There is a similar work from Dasari et al., ICRA 2023 [1], which is not cited but proposes very similar ideas. In [3], the authors also use human demos and train RL policies to track the human trajectories. There is also an alignment phase by planning the hand to the object (not learned). Diverse tasks and objects are also considered. I think BiDexHD differs by (1) learning the alignment and (2) distilling into vision-based policies; (1) is new but (2) is well-studied in previous work as the authors also agree. I see the existing ideas from [3] also address the motivation of designing unified and scalable framework for learning bimanual dexterous tasks.

There are no details about how the BC baseline is designed and trained. I imagine with an expressive enough policy parameterization, e.g., diffusion, the BC baseline can achieve nonzero success rates. I urge the authors to carefully provide the details of the BC experiments.

I also find the writing of the introduction section can be improved. The current form reads rather vague —"unified and scalable” is emphasized multiple times, which I understand by reading the approach section, but the introduction does not explain at all why the approach is unified and scalable. I think the statement of contributions can be greatly improved to provide more details about the overall approach.

[1] Learning Dexterous Manipulation from Exemplar Object Trajectories and Pre-Grasps, Sudeep Dasari, Abhinav Gupta, Vikash Kumar, ICRA 2023

**Questions:**

Can you comment on how you think of sim-to-real transfer of the setup? From the videos it seems the motion is quite unstable and jittery at times, do you think some kind of regularization or reward shaping can fix it? Or do you envision some fundamental challenge in sim-to-real?

Can you comment on why BiDexHD-IPPO underperforms in Test New (Table 1) compared to BiDexHD-PPO?

What is the batch size used in IPPO/PPO? I don’t see it listed in the appendix. I am curious about the effect of batch size on training stability.

It would be good to discuss how the success rates vary among tasks (Appendix C) also in the main text.

I suggest using a different notation, e.g., M_1 and M_2, instead of r_1 and r_2, for the two metrics in the experiment section since readers might mistake them as rewards.

---

> ### Author Response · Authors · 2024-11-22
> **Reply for Reviewer MLbc**
>
> Thanks for the detailed comments and valuable feedback! We are glad to address your questions and concerns one by one.
>
> `Q1`: "Technical contribution over existing work"
>
> `A1`: We thank the reviewer for detailedly comparing the difference in methodology between BiDexHD and PGDM [1]. In  `Author Rebuttal 2`, we list three major differences between PGDM and BiDexHD. To sum up, BiDexHD distinguishes existing work primarily in two aspects: "automatic task construction" and "general reward function".
>
> `Q2`: "Details of the BC experiments"
>
> `A2`: We have supplemented the configurations, architecture, and training details of the BC baseline in `Author Rebuttal 3` and `Appendix B.5`.  We analyze that **data quality and quantity** accounts for the bad performance of BC, regardless of Diffusion or ACT training methods. Please refer to `Author Rebuttal 3` and `Appendix B.5` for detailed explanation and demonstrations of the BC baseline on our project page [BiDexHD](https://sites.google.com/view/bidexhd).
>
> `Q3`: "Statement of contributions in the introduction"
>
> `A3`: We thank the reviewer for useful suggestions on improving the clarity of the contributions. We have emphasized the two major points in the red part `in the Introduction` with minor modifications.
>
> `Q4`: "Sim-to-real transfer"
>
> `A4`: In `Author Rebuttal 4` we explain the possible reasons for jerky motions demonstrated in the video and mention adding regularization terms to penalize joint angles, velocities, accelerations, and jerk to the ultimate objective is hopefully better to train a smoother policy. In `Author Rebuttal 5` we consider several major challenges and feasible solutions for real-world deployment of BiDexHD.
>
> `Q5`: "Performance comparison between BiDexHD-IPPO and BiDexHD-PPO in Test New tasks"
>
> `A5`: We explain the difference in detail in `Author Rebuttal 6`. Of course, both BiDexHD-IPPO and BiDexHD-PPO do not perform satisfactorily in Test New tasks, and it is somehow related to randomness as the number of Test New tasks is far less than trained tasks. BiDexHD-IPPO outperforms BiDexHD-PPO in most cases.
>
> `Q6`: "Batch size"
>
> `A6`: We thank the reviewer for pointing out that. We mention in `Appendix B.7` that our codebase is built upon UniDexGrasp++ [2]. The mini-batch size of IPPO / PPO / BC is 32 and a batch contains 3 minibatch, which is also commonly seen in other codebases. We have supplemented it `in Tables 5 & 6`.
>
> `Q7`: "Discussion about the success rates among tasks"
>
> `A7`: We thank the reviewer for the valuable suggestions. The red highlighted parts `in Sections 5.2 & 5.4` show some updated descriptions.
>
> `Q8`: "Notation for metrics"
>
> `A8`: We thank the reviewer for the substitution advice. To avoid causing further confusion for other reviewers, we will temporarily keep the notation $r_1,r_2$ and make this substitution in the final version of this paper!
>
> Thanks again for the review! We will implement the feedback in the final version of this paper. Further comments are welcome!
>
> **Reference**
>
> [1] Dasari, Sudeep, Abhinav Gupta, and Vikash Kumar. "Learning dexterous manipulation from exemplar object trajectories and pre-grasps." *ICRA 2023*.
>
> [2] Wan, Weikang, et al. "Unidexgrasp++: Improving dexterous grasping policy learning via geometry-aware curriculum and iterative generalist-specialist learning." *CVPR 2023*.

---

> ### Comment · Reviewer_MLbc · 2024-11-24
> **Reviewer response**
>
> I thank the author for answering my questions and concerns. I think the paper presentation in the revision has been improved significantly, and I raise the presentation and soundness score from 2 to 3. I don't have further concerns except for the comparison to PGDM.
>
> Re: application scope. I acknowledge that this work is more extensive in terms of tasks and settings considered, especially given this is a bimanual setup. I appreciate this but I don't think this sufficiently constitutes an acceptable submission, especially given the sim2real challenges.
>
> Re: Stage Division. I am still not convinced that BiDexHD differs from PGDM much --- I certainly acknowledge that BiDexHD learns the alignment and PGDM uses planning, but I am not convinced how the two approaches differ in empirical performance in practice. I might also argue that the simpler setup in PGDM circumvents carefully designing the reward functions. If you would like to claim significant contribution or novelty on this, I suggest comparing the learning approach to the planning approach in experiments. Of course this is challenging in two days. If you have more convincing arguments, or some illustrations supporting your claims, I would love to hear/see it and I am happy to discuss further.
>
> Re: Scalability. I mostly agree. PGDM does not require hand trajectory after pre-grasp though. I suggest adding discussions on this in the paper.

---

> > ### Author Response · Authors · 2024-11-25
> > **Further explanations for Reviewer MLbc**
> >
> > Thank you again for your thoughtful comments! We appreciate your detailed feedback and are happy to provide further explanations to address any remaining concerns.
> >
> > **Re: Significance (Application Scope):**
> > In this submission, we have extended BiDexHD to encompass various bimanual setups, including **diverse tool-object manipulation tasks and collaborative tasks involving a single object**. With its unified and scalable framework, and the anticipated availability of more high-quality Mocap data in the future, we believe BiDexHD is well-positioned to cover a wide range of bimanual rigid-body manipulation scenarios. This advancement is both meaningful and promising, laying the foundation for a bimanual generalist within the embodied AI community. To our knowledge, task diversity and scalability are the primary challenges in bimanual manipulation. By addressing these challenges, BiDexHD represents a **preliminary yet significant step toward scalable bimanual skill learning through multi-task reinforcement learning and vision policy distillation**.
> >
> > Regarding real-world deployment, we acknowledge the substantial challenges in the field, such as **simulation-to-reality gaps, control disparities, and safety concerns**. **It deserves more efforts to work on this** **through system design, algorithm modification, policy finetuning, and further debugging, .etc**. In `Author Rebuttal 5`, we have proposed feasible solutions to address the sim-to-real gap. And based on the current configurations, we are confident that BiDexHD is well positioned for future real-world deployment. Unlike some previous work [1] in which the dexterous hand floats in the air and can move freely, BiDexHD mounts bimanual hands onto robotic arms, adhering to the standard setup of modern robotics studies.
> >
> > **Re: Comparison (Stage Division & Scalability vs. PGDM):**
> > We would like to emphasize that **the core contributions of BiDexHD lie in (1) automatically constructing diverse bimanual tasks from human demonstrations without task-specific design and (2) solving them using a general reward function in a unified framework.** Based on these contributions, BiDexHD contrapuntally incorporates "alignment" and "tracking" stages, specifically designed for **more general, contact-rich behaviors**. For instance, as illustrated in Fig. 7 of `Appendix E`, in the alignment stage of the (empty, teapot, plate) task, the right hand must approach, grasp, re-orient, and lift the teapot and the left hand approaches and pushes the plate. Even with pre-grasp poses, achieving this intricate hand-object interaction purely through planning is challenging. This distinction highlights the difference between BiDexHD and PGDM. Additionally, beyond the data requirements (e.g., pre-grasp poses) and differences in setup mentioned previously, BiDexHD fundamentally diverges from PGDM in both task design and methodology. For a detailed comparison, including reorganized explanations and illustrations, please refer to the **updated section** in `Appendix E`.
> >
> > Considering that we have addressed most of your questions and concerns, would you mind raising your score in light of the updated information? We are sincerely grateful for your time and consideration. Thank you once again for your valuable feedback!

---

> > > ### Comment · Reviewer_MLbc · 2024-11-25
> > > **Response to the authors**
> > >
> > > I thank the authors for the detailed responses especially the added discussions in Appendix E. I appreciate the new Figure 7 illustrating the learned approach and reorientation behavior from BiDexHD.
> > >
> > > But again, I am still not convinced that this learning approach, despite being different from PGDM, would work fundamentally better than PGDM's planning approach. In the example in Figure 7, we can specify a pre-grasp of the object while it is on the table, and then specify the trajectory of the object being re-oriented and lifted (and the rest) --- the approach part can be planned, and the rest including reorientation and lifting can be learned using trajectory tracking reward. I disagree with the comment that "It is hard to realize this hand-object interaction only through planning-based methods like PGDM".
> > >
> > > Do you have any comment on this? One argument might be that providing the (hand-)object trajectory of reorienting and lifting would be very difficult so BiDexHD is more scalable, but I don't agree with that for now.
> > >
> > > Disclaimer: I am not an author of PGDM. I worked on the similar problem before.

---

> > > > ### Author Response · Authors · 2024-11-26
> > > > **Further explanations for Reviewer MLbc**
> > > >
> > > > Thanks for your reply! We would like to address your remaining concerns.
> > > >
> > > > It is important to highlight that in BiDexHD, **object trajectories are not provided during the alignment stage**. As illustrated in **updated Fig. 7** in `Appendix E`, object trajectory tracking starts only after the simulation-dataset alignment has been successfully completed. Namely, the state described in the 4th frame is aligned with the initial state of the demonstrated trajectory. Taking the (empty, teapot, plate) task as an example, the demonstrated trajectory starts at the state where "the left hand pushes the plate to pose C, and the right hand moves the teapot to pose D." No additional trajectory information is provided before this state in the dataset. In this context, BiDexHD allows us to (1) initialize the object-tool pair at any pose (i.e., poses A and B can be randomized) in the simulation. Through reinforcement learning, we can learn contact-rich behaviors—such as moving, grasping, twisting, and pushing—that successfully relocate the object-tool pair to their desired poses C and D, thus aligning with the initial state of the demonstrated trajectory. (2) It is nearly impossible to specify additional task-specific guidance for the simulation-dataset alignment across diverse tasks, and doing so would contradict one of BiDexHD's core principles: "using a general approach to solve all constructed bimanual tasks."
> > > >
> > > > Furthermore, we would like to reiterate the two core claims of BiDexHD: **"automatic task construction" and "general reward function."** We present BiDexHD as a unified bimanual framework with significant potential for large-scale extension. Although it may seem similar to PGDM in some aspects, we have specifically designed distinct learning strategies to support these claims.
> > > >
> > > > Thanks again for the review! If most of your questions and concerns are addressed, would you mind raising your score? We are sincerely grateful for your time and consideration.

---

> ### Comment · Reviewer_MLbc · 2024-11-26
> **Reviewer response**
>
> Thanks for the clarifications. I understand that object trajectories are not provided during the alignment stage.
>
> My concern is, it seems to me that providing the object trajectories *before* the alignment stage, i.e., the object being re-oriented and lifted, is rather trivial? Suppose we have such trajectories, then planning to the pre-grasp and then learning the rest with trajectory tracking reward would also work.
>
> I totally agree with the core philosophies of "automatic task construction" and "general reward function.", but it seems to me that prior work like PGDM also does that, by converting an object trajectory, which is slightly more involved than the one BiDexHD requires, into a task specification with the tracking reward. Fundamentally, I don't think the two approaches differ much, if the object trajectory being lifted and re-oriented can also be provided, which seems to me, rather easy.
>
> Also, I hope this is clear from my previous comments: if the paper solves the general sim2real setting, I will be very supportive of the paper acceptance. However, since that is not the case, I would like to see more novelty and technical contribution of the method itself. Currently I am not yet convinced that the paper proposes a new approach that addresses the shortcoming of previous work.

---

> > ### Author Response · Authors · 2024-11-26
> > **Further rebuttal for Reviewer MLbc**
> >
> > Thank you for your reply! We would like to summarize the major contributions of BiDexHD in comparison to **PGDM** below.
> >
> > - PGDM primarily focuses on **single Adroit Hand grasping in Mujoco** simulations, while BiDexHD extends to more complex **bimanual arm-hand systems and manipulation tasks in highly parallelized IsaacGym** simulations.
> > - PGDM derives pre-grasp poses from sources such as MoCap, Tele-Op, human labels, or learned models, which inherently **introduce significant human effort**. In contrast, BiDexHD is more general and scalable, as it **does not require any additional annotations**.
> > - PGDM mainly targets **single-hand grasping tasks**, whereas BiDexHD tackles a broader range of bimanual tasks beyond grasping, such as stabilizing a heavy box or pushing a plate. For example, in the (empty, teapot, plate) task shown in Fig. 7, where the plate is randomly initialized, the first step involves pushing it to the reference pose. In the case of planning-based methods, a critical question arises: **How can we infer the “pre-grasp” pose, when the task does not involve grasping?** BiDexHD does not rely on prior poses and can effectively adapt to diverse bimanual manipulation tasks.
> > - PGDM reaches a target object through planning and learns to grasp and move it along a specific trajectory via reinforcement learning. BiDexHD employs unified reinforcement learning, starting with aligning both hands and objects with a reference state followed by object trajectory tracking. This design enables BiDexHD to learn various **contact-rich behaviors**, such as twisting and pushing, beyond simple reaching and grasping, while **maintaining a balance between the quality of policy and training difficulty**.
> > - In PGDM, given a single human demonstration $\tau$, the model is **limited to solving tasks where objects are initialized ideally to the initialization of $\tau$**, whereas BiDexHD can learn these tasks with **arbitrarily initialized objects** through end-to-end RL, utilizing a generally designed two-stage reward.
> >
> > Considering these key differences, we believe BiDexHD is a novel and distinctive work compared to PGDM, as it **addresses harder tasks, loosens data requirements, and demonstrates better scalability and adaptation capabilities**.
> >
> > Thank you again for the review! If most of your questions and concerns have been addressed, would you mind raising your score? We sincerely appreciate your time and consideration.

---

> > > ### Author Response · Authors · 2024-11-28
> > > **For Reviewer MLbc**
> > >
> > > Here, we respond to your comments about **the comparison with previous work** and address the issues. If our rebuttal has addressed your concerns, we would be grateful if you would **kindly consider revising your score in response**. If you have further questions, feel **free to let us know**. We hope to hear back from you!

---

### Official Review · Reviewer_WD9o · 2024-11-03

**Soundness:** 3
**Presentation:** 3
**Contribution:** 2
**Rating:** 6
**Confidence:** 5

**Summary:**

This paper introduces **BiDexHD**, a framework designed to automatically turn a human bimanual manipulation dataset into simulation tasks and learn diverse bimanual dexterous manipulation skills with teacher-student method (RL in sim with state-based observations + distill into a policy with point cloud observation via Dagger). It aims to address the complexity and lack of task diversity in existing approaches. The authors propose a novel frame that:

1. instead of manually designed or predefined tasks, it creates feasible tasks from given bimanual trajectories;
2. using a teacher-student learning framework leveraging unified two-staged reward functions and vision-based policy distillation
3. evaluated on the TACO dataset across 141 tasks with strong performance on both seen and unseen tasks

The central claim of this paper is centered around the notion of "a preliminary attempt towards universal bimanual skills", and a framework that is "unified and scalable".

As in its current form, I cannot argue for its acceptance, due to the questions and weaknesses enumerated below. There is not enough evidence to support the strong claims of this work.

**Strengths:**

Originality:
- The framework described in the paper is novel in these perspectives:
1. While it is built on the widely used teacher-student framework, the authors designed two-staged reward functions that are universal to tool-use bimanual manipulation tasks constructed in simulation.
2. The idea of constructing many tasks from given bimanual trajectories is also a creative way of addressing the scalability challenge in simulation.

Quality:
- The authors performed several ablation experiments to demonstrate the effectiveness of each design choice, ranging from IPPO vs PPO, thresholds used in reward, each stage of the reward/training, functional grasping center, future conditioning steps, and baseline methods (BC). These efforts improved the technical soundness of the proposed method.

Clarity:
- I appreciate the clarity of the writing in terms of explaining the stages in this framework, the visual illustrations of each stage in figure 1 and figure 2, and the explanations for reward functions.

Significance:
- The problem of bimanual manipulation is becoming increasingly important for general-purpose robot intelligence. As the author pointed out, human-level performance on challenging bimanual dexterous manipulation skills is crucial for tasks involving coordination and tool usage. This work attempts to provide a unified and scalable framework that addresses several constraints in prior works.

**Weaknesses:**

1. This work limits the training and testing to one dataset and one simulator for a method that aims to be universal and scalable. If it is feasible, a demonstration of the proposed framework's effectiveness in other bimanual manipulation benchmarks and simulation environments would provide more convincing signals.

2. While there are 141 tasks from 6 categories, they are limited to tool-usage-oriented tasks, and the reward functions are tailored for solving this flavor of tasks. However, bimanual manipulation tasks that do not fall into this genre would likely require separate reward function designs. For example, cloth folding, packing & unpacking, or assembling & disassembling tasks are harder to simulate and have well-defined reward functions. As is commonly known, designing and tuning a good reward function for RL requires human effort and knowledge, which are both demanding and challenging. Thus, further experimental designs and evaluations are needed to strengthen the scalable and universal claim.

2. It is confusing that the BC baseline trained using teleoperated data achieves 0 success even on trained tasks. Could authors include more details on BC data's quality + quantity, training, and evaluation setting? To the best of my knowledge, other works have shown non-zero success for BC methods (even as baselines) on bimanual dexterous manipulation tasks.[1]

[1] Towards Human-Level Bimanual Dexterous Manipulation with Reinforcement Learning, Chen et al.

**Questions:**

1. The dataset used in this paper to generate tasks and provide trajectories is TACO: Benchmarking Generalizable Bimanual
Tool-ACtion-Object Understanding. In this dataset, there are 2.5k sequences of precise hand-object meshes and annotations. However, high-quality datasets such as TACO are costly to collect and difficult to scale as they require motion-tracking equipment and facilities. My questions are: how would reward engineering, depending on privileged state information in simulation tasks, that are constructed from motion-tracked trajectories, scale to infinitely diverse bimanual dexterous manipulation tasks? What are the limitations and bottlenecks?

2. Although I understand this work pertains to the study of bimanual dexterous manipulation framework in simulation, what would be some practical bottlenecks preventing this framework from being deployed onto a real bimanual dexterous robot?

3. For automatically constructed tasks in simulation, what are some limitations to the current framework preventing it from being "easily" scalable or applicable to other tasks? Or what quantifies as "easily"? (as the word "easily" is used in the paper)

---

> ### Author Response · Authors · 2024-11-22
> **Reply for Reviewer WD9o [1/2]**
>
> Thanks for the detailed comments and insightful review! We are encouraged that the reviewer shows a positive attitude towards our writing, illustrations, and the core idea of "constructing tasks from given bimanual trajectories to address the scalability challenge". We are glad to provide a point-by-point response below.
>
> `Q1`: "Only one dataset"
>
> `A1`: We have extended our BiDexHD to a new bimanual dataset Arctic. Four cooperative tasks of a single object show that our unified framework is scalable and transferable to different types of bimanual tasks and datasets. Please refer to `Author Rebuttal 1` for descriptions,  `Appendix B.6` for details, and our website page [BiDexHD](https://sites.google.com/view/bidexhd) (in the second to last section) for video demonstrations.
>
> `Q2`: "More types of tasks to support the scalable and universal claim"
>
> `A2`: In this submission, we primarily focus on bimanual rigid-body manipulation tasks, including bimanual tool-usage-oriented and collaborative tasks. With the BiDexHD framework, we can efficiently scale up using a generally designed two-stage reward function to address these types of tasks. We appreciate the reviewer’s insightful observation that tasks such as cloth folding, packing and unpacking, and assembling and disassembling represent more challenging bimanual collaborative scenarios due to their simulation complexity. From another perspective, these tasks encompass distinct categories of manipulation challenges: cloth folding exemplifies soft-object manipulation, where object coordinates are difficult to stabilize; packing involves articulated-object manipulation, requiring the specification of object articulation; and assembling highlights precise robotic manipulation. These categories fall outside the scope of this paper. We position BiDexHD as a unified framework tailored to bimanual rigid-body-centric manipulation tasks, which are primarily concerned with object pose transformations. However, we are excited about the potential to extend this framework to incorporate a broader range of manipulation tasks in future work.
>
> `Q3`: "BC baseline"
>
> `A3`: Since each task in BiDexHD is built from a single demonstration, we do behavior cloning from a single retargeted observation-action sequence. All the training and evaluation configurations match the student vision-based policy learning. We display some demonstrations of the BC baseline on our project page [BiDexHD](https://sites.google.com/view/bidexhd) and provide a detailed analysis in `Author Rebuttal 3` and  `Appendix B.5`.
>
> `Q4`: "Limitations and bottlenecks about scaling with Mocap trajectories"
>
> `A4`: We believe the most prominent bottleneck at present is the limited availability of high-quality demonstrations that are both temporally aligned and physically aware. This challenge arises partly from the difficulty of precise hand and object pose detection from raw vision signals, as well as the labor-intensive and time-consuming nature of data collection processes. However, we are confident that quality and quantity will not remain obstacles in the future. In fact, for BiDexHD, we intentionally selected Mocap data as the source because Mocap systems are relatively lighter, more cost-effective, and portable compared to more complex leader-follower systems [1] or exoskeleton systems [2]. In comparison, Mocap holds significant potential for scalability. BiDexHD is such a general framework designed to seamlessly adapt to so many existing high-quality Mocap human bimanual datasets, such as AKINK [3] and [4]. And it is well-positioned to scale further with data from advanced Mocap systems, such as [5], in the future.
>
> `Q5`: "Bottlenecks of real-world deployment"
>
> `A5`: We detailedly analyze the vision gap, controller gap, physics (simulation) gap, and safety concerns in `Author Rebuttal 5`. We will deploy our policy with proper modifications to real bimanual systems in the future.
>
> `Q6`: "Limitations to the current framework and explanation to easily"
>
> `A6`: We detailedly survey the limitations of current studies and compare the major differences of related work in `Author Rebuttal 2`. The most major points can be summarized as "automatic task construction" and "general reward function". We define "easy" as minimal efforts on code modifications, dataset preprocessing, training and evaluation, configurations and hyperparameters, .etc. As a unified framework, we believe BiDexHD naturally owns this property.
>
> Thanks again for the review! We will implement the feedback in the final version of this paper. Further comments are welcome!

---

> > ### Author Response · Authors · 2024-11-22
> > **Reply for Reviewer WD9o [2/2]**
> >
> > **Reference**
> >
> > [1] Zhao, Tony Z., et al. "Learning fine-grained bimanual manipulation with low-cost hardware." *ArXiv 2023*.
> >
> > [2] Fang, Hongjie, et al. "Airexo: Low-cost exoskeletons for learning whole-arm manipulation in the wild." *ICRA 2024*.
> >
> > [3] Zhan, Xinyu, et al. "OAKINK2: A Dataset of Bimanual Hands-Object Manipulation in Complex Task Completion." *CVPR 2024*.
> >
> > [4] Razali, Haziq, and Yiannis Demiris. "Action-conditioned generation of bimanual object manipulation sequences." *AAAI 2023*.
> >
> > [5] Wang, Chen, et al. "Dexcap: Scalable and portable mocap data collection system for dexterous manipulation." *ArXiv 2024*.

---

> ### Author Response · Authors · 2024-11-25
> **For Reviewer WD9o**
>
> Thanks again for your careful review! Here, we respond to your comments and address the issues. We hope to hear back from you! If you have further questions, feel free to let us know, and ***\*we are more than happy to answer additional questions\****. If you feel that our rebuttal has addressed your concerns, we would be grateful if you would consider ***\*revising your score in response\****.

---

> ### Comment · Reviewer_WD9o · 2024-11-26
> **Reply to the authors**
>
> Thank you for the detailed response to my questions and concerns. I truly appreciate the effort to add additional datasets and tasks.
>
> The proposed method uses real-world mocap data, constructs simulation tasks automatically based on that, and then trains RL agents with a universal reward function in simulation to solve these tasks. The only piece missing from this work is the "to real" component. This would make it impactful as contemporary real2real imitation learning (ACT, UMI, etc) or reinforcement learning work (SERL), or sim2real work (a bunch of dex hand manipulation work), or more recently real2sim2real works. Hopefully, the authors could research this in the future.
>
> While the limitations to the method proposed in this paper still remain, i.e. it depends on mocap datasets and the automatic task generation pipeline "primarily concerned with object pose transformations.", weakening the "general" and "unified" claim to large bodies of bimanual manipulation challenges, the proposed method is indeed automatic and universal to the problems that are studied in this paper. The experiments and new info from the rebuttal and appendix provide enough evidence to support this. Thus, I would increase the soundness score to 3 and slightly increase the recommendation.

---

> > ### Author Response · Authors · 2024-11-26
> > **For Reviewer WD9o**
> >
> > We sincerely appreciate Reviewer WD9o for revising your score and acknowledging that our current submission provides sufficient evidence to support the "general" and "unified" claims. Regarding your remaining concerns: (1) real experiments, and (2) expanding to more types of bimanual manipulation tasks, we are actively working on deploying BiDexHD on real bimanual robotic systems and extending it to a broader range of tasks. We hope to present promising results in the final revision of the paper. Once again, thank you for your thoughtful review!

---

### Author Response · Authors · 2024-11-22
**Author Rebuttal for Common Questions [1/6]**

We sincerely thank the four reviewers for their thoughtful comments! We have completed some supplements according to reviewers' suggestions, and we summarize the major changes as follows. All modifications in the revised paper are marked red and all these supplements will be incorporated into the final version of this paper.

- We have extended our BiDexHD framework to a new bimanual dataset **Arctic** [1], which mainly focuses on bimanual cooperative tasks of a single object. The results demonstrate that our BiDexHD is scalable and transferable to different types of bimanual tasks and datasets. We have supplemented the detailed descriptions in `Appendix B.6` and displayed the video demonstrations on our website page [BiDexHD](https://sites.google.com/view/bidexhd) (in the second to last section).
- We have supplemented the **configurations, architecture, and training details** of the BC baseline in `Appendix B.5` and added the **video demonstrations** of BC (showing how this baseline fails) on our website page [BiDexHD](https://sites.google.com/view/bidexhd) (in the last section).

We provide more detailed explanations as follows:

1. About **task diversity**. We thank the reviewers for pointing out that more categories of tasks need to be involved to prove the scalability and generalizability of BiDexHD. Considering we primarily focus on diverse bimanual manipulation tasks in this paper,  we extend our framework to a popular bimanual dataset Arctic focusing on bimanual tasks of a single object. We build up four tasks `Mixer Holding, Capsule Machine Grabbing, Box Flipping, and Ketchup Lifting` from four trajectories in the Arctic dataset and follow the pipeline of teacher learning to learn a state-based policy for each task. The average success rate of stage one $r_1$ and trajectory tracking rate $r_2$ shown in the table below demonstrate the effectiveness and generalizability of BiDexHD in collaborative bimanual manipulation tasks. We further visualize the behavior of tasks from the Arctic dataset on our page [BiDexHD](https://sites.google.com/view/bidexhd) (in the last section).

|           Task           | Train $r_1$(\%) | Train $r_2$(\%) |
| :----------------------| :-------------: | :-------------: |
|      Mixer Holding       |      90.01      |      79.24      |
| Capsule Machine Grabbing |      96.47      |      93.45      |
|       Box Flipping       |      94.10      |      91.23      |
|     Ketchup Lifting      |      93.98      |      82.99      |

---

### Author Response · Authors · 2024-11-22
**Author Rebuttal for Common Questions [2/6]**

2. About **comparison with other bimanual studies**.

  The majority of existing studies focusing on bimanual manipulation exhibit two features:

  - They are limited to a certain category of tasks or existing benchmarks with a limited range of tasks.
  - For RL-based methods, they tailor specific reward functions to specific tasks. For IL-based methods, it is inevitable to collect a bulk of data for learning specific tasks (typically around 50 trajectories for each single task).

  We would like to emphasize that **BiDexHD is the first framework to (1) automatically construct diverse bimanual tasks from human demonstrations without task-specific design, and (2) solve them using a general reward function in a unified manner**. This unique feature enables the framework to potentially scale to an infinite variety of bimanual dexterous manipulation tasks, given sufficient datasets. It paves the way toward developing a generalizable foundation policy through distillation.

  We appreciate the reviewers for bringing up a similar work, PGDM. However, there are significant differences between PGDM and BiDexHD:

  - **Stage Division**:

    - PGDM divides the task into distinct stages: planning a pre-grasp pose (reaching stage) and learning to grasp and move through reinforcement learning (grasping and moving stages). Their planning-based reaching stage is limited to performing hand-reaching behaviors, while our BiDexHD can perform general, contact-rich behaviors in the RL-based alignment stage.

    - BiDexHD employs unified reinforcement learning, starting with aligning both hands and objects to a ready state (alignment stage), followed by trajectory tracking (tracking stage). This design allows BiDexHD to flexibly learn diverse skills like twisting and pushing, going beyond simple reaching and grasping. Once both hands securely hold the objects, they maintain their relative states and learn to track desired poses with ease. Our design properly strikes a balance between policy quality and training difficulty.

  - **Scalability**:
    - PGDM relies on human-annotated pre-grasp poses in the TCDM benchmark, limiting its applicability to broader tasks.
    - BiDexHD only requires a pair of tool-object trajectories from a dataset for each task along with a reference hand pose to calculate the grasping center without extra annotations, enhancing its scalability.
  - **Application Scope**:
    - PGDM primarily focuses on single Adroit Hand manipulation in Mujoco simulations.
    - BiDexHD extends to more complex bimanual arm-hand systems in highly parallelized IsaacGym simulations.

  We thank the reviewers for mentioning another recent bimanual work DexCap [3] which proposes a novel motion capture and vision-based data collection system for bimanual task learning via imitation. However, **their collected data alone is insufficient to derive feasible policies**, necessitating further human-in-the-loop finetuning. In contrast, BiDexHD uses online reinforcement learning with a general reward function to learn diverse bimanual skills from object motion capture data through trial and error, without additional fine-tuning.

---

### Author Response · Authors · 2024-11-22
**Author Rebuttal for Common Questions [3/6]**

3. About the **BC baseline**. To get the arm and hand action labels for imitation learning, we employ Dexpilot to retarget human hand motions in the TACO dataset to hand joint angles for dexterous hands and solve inverse kinematics (IK) to convert Mocap 6D wrist pose to 6-DOF arm joint angles. Since each task is built from a single demonstration, we adopt vanilla imitation learning to directly learn a vision-based policy $\pi _ \phi^\text{side}(\mathbf{a} _ t^\text{side}|\mathbf{o} _ t^\text{side},\mathbf{a}  _  {t-1}^\text{side}),\mathbf{o} _ t=[(\mathbf{j}, \mathbf{v})^\text{side},(\mathbf{x},\mathbf{q})^\text{side,w},\mathbf{x}^{\text{side,ft}},\text{pc}^\text{obj}] _ t$ for each task from a single observation-action sequence after retargeting. The loss function is the standard MSE loss.  Experimental results show that imitation learning from a single trajectory fails. We visualize some demonstrations of the BC baseline on our project page [BiDexHD](https://sites.google.com/view/bidexhd). We analyze the primary reasons for these failures are:

- **Limited Demonstrations**: Only one demonstration is available for imitation learning, leaving large portions of the observation space unexplored. As a result, BC struggles with unvisited states, due to distribution shift.
- **Mismatched Kinematics & Dynamics**. Though robot trajectories derived from retargeting seem to be aligned with human demonstrations spatially and temporally, they exhibit inconsistent kinematics and unreal dynamics. Therefore, the retargeted trajectories are **non-expert**, not satisfying the quality requirements for BC. This results in fragile policies prone to failure as shown in the videos on the page.

In contrast, existing practices in IL-based bimanual manipulation usually require **$20\sim 50$ high-quality teleoperation data (not retargeted human data)** per task. In conclusion, **data quality and quantity** account for the bad performance of BC.

---

### Author Response · Authors · 2024-11-22
**Author Rebuttal for Common Questions [4/6]**

4. About **jerky motion**. We would like to provide some explanations to the point that the motions demonstrated in the videos appear "not so smooth".

  - When recording these videos in IsaacGym, we did not insert `time.sleep(control_dt)` between consecutive steps, causing the rendered frames to play faster than the actual execution. As a result, the recorded videos run at approximately three times the intended speed.
  - Furthermore, we position BiDexHD as a preliminary attempt towards scalable bimanual skill learning from diverse constructed tasks. Therefore, this submission **prioritizes achieving high task completion rates** for challenging bimanual dexterous tasks in simulation. Smoother motions for sim-to-real deployment can be achieved by adding regularization terms to penalize joint angles, velocities, accelerations, and jerks. These have been common practices in previous work. Future work could incorporate these improvements for sim-to-real deployment.

---

### Author Response · Authors · 2024-11-22
**Author Rebuttal for Common Questions [5/6]**

5. About **real-world deployment**. We consider real-world deployment of bimanual systems may face several challenges:

  - **Vision Gap**: The point clouds synthesized from RGBD frames are noisy.
  - **Controller Gap**: Though both simulation and real-world robots can apply joint position control using the PD controller, the hardware controller cannot perfectly match the simulated controller.
  - **Physics (Simulation) Gap**: For contact-rich bimanual manipulation tasks, IsaacGym can not perfectly simulate all complicated physical properties and dynamics for the interaction between robots and objects.
  - **Safety and Reliability**: Since we focus on tabletop tasks, collisions (self-collision / collision between robots and the table) could cause damage in a real-world deployment.

  To deploy our trained vision-based policy to real robots, we could address these challenges in future work:

  - Various types of randomizations on point clouds should be included in the simulation to bridge the visual gap.
  - We should include domain randomization, including randomization on all objects' states, parameters of the controller, external forces, and physical properties of rigid bodies, to bridge the controller and physics gap.
  - For safety concerns, we should include action penalty terms to the reward function for smoother policies and add some force sensors to mitigate collisions.

  Regarding the comment that "future positions of objects can not be easily incorporated in real-world experiments", we should first clarify that these trajectories of objects are necessary **task plans** that tell what the agent should do. For example, when a cup is provided, the plan of the cup's trajectory specifies whether to pour the water or place it somewhere. Without such a plan, the task will be ambiguous. These trajectories should be generated by the high-level task planning models, while BiDexHD focuses on low-level close-loop control. Recent works [5,6] have studied some feasible solutions for object trajectory planning:

  - Use **large multimodal models** to generate valid future trajectories according to historical observations and trajectories.
  - Train an **object motion prediction model** from various object manipulation datasets.

---

### Author Response · Authors · 2024-11-22
**Author Rebuttal for Common Questions [6/6]**

6. About **the performance difference between BiDex variants**. Learning multi-finger dexterous manipulation policy with high-dimension action space is inherently challenging for reinforcement learning from scratch. As claimed in our paper, considering that each hand has its object focus, within equal limited (~10k minibatch) PPO updates, BiDexHD-IPPO is more efficient in terms of single-hand policy learning than centralized PPO. Therefore, the overall average results demonstrate the superiority of BiDexHD-IPPO over BiDexHD-PPO. In Test Combinational tasks the objects all come from the training set and either hand has learned the manipulation skill, so it is more possible for BiDexHD-IPPO to adapt to these tasks because BiDexHD-IPPO trains independent expert policies focusing solely on specific groups of objects. However, in Test New tasks BiDexHD-IPPO loses the advantage, and the empirical results show that BiDexHD-PPO, jointly attending to both hands and objects, performs slightly better. Both variants are part of the BiDexHD framework, and we are glad to incorporate more RL variants like multi-agent algorithms in future work.

**Reference**

[1] Fan, Zicong, et al. "ARCTIC: A dataset for dexterous bimanual hand-object manipulation." *CVPR 2023*.

[2] Dasari, Sudeep, Abhinav Gupta, and Vikash Kumar. "Learning dexterous manipulation from exemplar object trajectories and pre-grasps." *ICRA 2023*.

[3] Wang, Chen, et al. "Dexcap: Scalable and portable mocap data collection system for dexterous manipulation." *ArXiv 2024*.

[4] Wen, Bowen, et al. "Foundationpose: Unified 6d pose estimation and tracking of novel objects." *CVPR 2024*.

[5] Wang, Ye, et al. "Quo Vadis, Motion Generation? From Large Language Models to Large Motion Models." *ArXiv 2024*.

[6] Chen, Yuanpei, et al. "Object-Centric Dexterous Manipulation from Human Motion Data." *ArXiv 2024*.

---

### Meta-Review · Area_Chair_gc7T · 2024-12-20

**Metareview:**

The submission introduces a framework for learning bimanual dexterous manipulation skills by leveraging human demonstrations. The reviewers acknowledge its novelty in task construction and the teacher-student framework, as well as its promising performance on the TACO dataset. However, they highlight several areas for improvement and concerns regarding its universality, scalability, and novelty compared to existing work.

Strengths:

The paper presents a novel approach to automatic task construction from human demonstrations and a unified reward function for reinforcement learning, addressing a significant challenge in bimanual dexterous manipulation. The empirical results are robust, showcasing competitive task fulfillment rates, including zero-shot generalization.

Weaknesses:

Several reviewers raised concerns about the limited scope of the experiments (one dataset and simulation environment), questioning claims of universality and scalability. Additionally, the BC baseline setup and performance metrics were criticized as unclear or inadequate. Concerns about novelty over existing methods like PGDM were also frequently mentioned.

While the paper presents promising results and tackles an important problem, the concerns about novelty, scalability, and universality outweigh its strengths in the current form. Therefore, I recommend rejection for this iteration.

**Additional Comments On Reviewer Discussion:**

The authors made substantial efforts to address reviewer concerns through detailed rebuttals and by providing additional results. Despite these efforts, some reviewers remain unconvinced about BiDexHD’s contributions compared to PGDM and its practical applicability, especially in sim-to-real settings.

---

### Decision · Program_Chairs · 2025-01-22

Reject